



# A global study of hygroscopicity-driven light scattering enhancement in the context of other in-situ aerosol optical properties

Gloria Titos[1,2], María A. Burgos[3], Paul Zieger[3], Lucas Alados-Arboledas[1,2], Urs Baltensperger[4], Anne Jefferson[5], James Sherman[6], Ernest Weingartner[4,7], Bas Henzing[8], Krista Luoma[9], Colin O'Dowd[10], Alfred Wiedensohler[11], and Elisabeth Andrews[5,12]

[1]Andalusian Institute for Earth System Research, University of Granada, 18006, Granada, Spain
[2]Department of Applied Physics, University of Granada, 18071, Granada, Spain
[3]Department of Environmental Science & Bolin Centre for Climate Research, Stockholm University, 11418, Stockholm, Sweden
[4]Laboratory of Atmospheric Chemistry, Paul Scherrer Institute, Villigen, Switzerland
[5]Cooperative Institute for Research in Environmental Studies, University of Colorado, 80309, Boulder, USA.
[6]Department of Physics and Astronomy, Appalachian State University, Boone, USA
[7]Now at: Institute for Sensing and Electronics, University of Applied Sciences, Windisch, Switzerland
[8]Netherlands Organisation for Applied Scientific Research (TNO), Princetonlaan 6, 3584 Utrecht, the Netherlands
[9]Institute for Atmospheric and Earth System Research/Physics, Faculty of Science, University of Helsinki, P.O. Box 68, 00014 Helsinki, Finland
[10]School of Physics, Ryan Institute's Centre for Climate and Air Pollution Studies, National University of Ireland Galway, University Road, H91 CF50 Galway, Ireland
[11]Leibniz Institute for Tropospheric Research, DE-04318 Leipzig, Germany
[12]Earth System Research Laboratory, National Oceanic and Atmospheric Administration, 80305, Boulder, USA

**Correspondence:** G. Titos (gtitos@ugr.es)

**Abstract.** The scattering and backscattering enhancement factors ($f$(RH) and $f_b$(RH)) describe how aerosol particle light scattering and backscattering, respectively, change with relative humidity (RH). They are important parameters in estimating direct aerosol radiative forcing (DARF). In this study we use the dataset presented in Burgos et al. (2019) that compiles $f$(RH) and $f_b$(RH) measurements at three wavelengths (i.e. 450, 550 and 700 nm) performed with tandem nephelometer systems

5 at multiple sites around the world. We present an overview of $f$(RH) and $f_b$(RH) based on both long-term and campaign observations from 23 sites representing a range of aerosol types. The scattering enhancement shows a strong variability from site to site, with no clear pattern with respect to total scattering coefficient. In general, higher $f$(RH) is observed at Arctic and marine sites while lower values are found at urban and desert sites, although a consistent pattern as a function of site type is not observed. The backscattering enhancement $f_b$(RH) is consistently lower than $f$(RH) at all sites, with the difference

10 between $f$(RH) and $f_b$(RH) increasing for aerosol with higher $f$(RH). This is consistent with Mie theory which predicts higher enhancement of the light-scattering in the forward than in the backward direction as the particle takes up water. Our results show that the scattering enhancement is higher for PM$_1$ than PM$_{10}$ at most sites, which is also supported by theory due to the change in scattering efficiency with the size parameter that relates particle size and wavelength of incident light. At marine-influenced sites this difference is enhanced when coarse particles (likely sea salt) predominate. For most sites, $f$(RH) is observed to

15 increase with increasing wavelength, except at sites with a known dust influence where the spectral dependence of $f$(RH) is





found to be low or even exhibit the opposite pattern. The impact of RH on aerosol properties used to calculate radiative forcing (e.g., single scattering albedo, $\omega_0$, and backscattered fraction, $b$) is evaluated. The single scattering albedo generally increases with RH while $b$ decreases. The net effect of aerosol hygroscopicity on radiative forcing efficiency (RFE) is an increase in the absolute forcing effect (negative sign) by a factor of up to 4 at RH=90% compared to dry conditions (RH<40%). Because of the scarcity of scattering enhancement measurements, an attempt was made to use other, more commonly available aerosol parameters (i.e., $\omega_0$ and scattering Angström exponent, $\alpha_{\mathrm{sp}}$) to parameterize $f$(RH). The majority of sites (75%) showed a consistent trend with $\omega_0$ (higher $f$(RH=85%) for higher $\omega_0$), while no clear pattern was observed between $f$(RH=85%) and $\alpha_{\mathrm{sp}}$. This suggests that aerosol $\omega_0$ is more promising than $\alpha_{\mathrm{sp}}$ as a surrogate for the scattering enhancement factor, although neither parameter is ideal. Nonetheless, the qualitative relationship observed between $\omega_0$ and $f$(RH) could serve as a constraint on global model simulations.

## 1 Introduction

Aerosol particles, both from natural and anthropogenic sources, interact with solar radiation through scattering and absorption. The direct aerosol radiative forcing (DARF) results from changes in the top of the atmosphere and surface net fluxes associated with the aerosol scattering and absorbing behavior. The optical properties of the aerosol particles largely govern the magnitude of their radiative impact. Thus, uncertainties in these properties contribute to uncertainties in aerosol radiative forcing and are also important for visibility estimates.

Depending on their size and chemical composition, aerosol particles are able to take up water and become larger in size than their dry equivalents. Water uptake by aerosols changes not only the particle size but also composition (reflected in the aerosol refractive index) and this impacts the magnitude and angular distribution of scattered light. Aerosol absorption may also be impacted by RH if absorbing aerosols become embedded in a scattering shell (Bond et al., 2006; Fuller et al., 1999). Thus, the magnitude of the DARF will be affected by aerosol hygroscopic growth. The uncertainty related to parameterizing aerosol water uptake may be one contributor to the large differences observed among global climate models (e.g., Burgos et al., 2020) when simulating the direct aerosol effect (Myhre et al., 2013; Boucher and Anderson, 1995; Curci et al., 2015).

The influence of aerosol hygroscopicity on the particle light scattering coefficient is usually quantified by means of the scattering enhancement factor, $f$(RH, $\lambda$), which is typically defined as the ratio of the scattering coefficient ($\sigma_{\mathrm{sp}}$) at some high relative humidity (RH) to the scattering coefficient at a low reference RH ($RH_{dry}$) (Covert et al., 1972), as shown in Eq. 1:

$$f(RH,\lambda) = \frac{\sigma_{sp}(RH,\lambda)}{\sigma_{sp}(RH_{dry},\lambda)}. \tag{1}$$

Hereafter, $f$(RH) refers to the 550 nm wavelength unless otherwise noted, with the wavelength dependence omitted for simplicity. Since the 1970s there have been multiple deployments of surface in-situ instrumentation to measure $f$(RH) of the atmospheric aerosol across a wide range of aerosol types. Climatological information on $f$(RH) can provide useful information about the diversity of the effect of aerosol hygroscopicity on light scattering. Titos et al. (2016) present a review of many of these previous observations and point out the need to use a harmonized dataset to perform a joint $f$(RH) analysis. Even




at individual sites, changes in emission sources and air masses impacting the sites can cause variability in observed $f$(RH)
(e.g., McInnes et al., 1998; Carrico et al., 2003; Zieger et al., 2013; Titos et al., 2014a). Assessing the underlying causes of the

diversity both spatially (i.e., across sites) and temporally (i.e., at a single site) is important, particularly if a goal is to constrain
Earth system model parameterizations.

As mentioned above, water uptake by aerosols can modify their angular scattering properties and therefore may also affect the
backscatter fraction (amount of radiation that is scattered in the backward direction compared with the total scatter radiation),
further influencing the DARF. The backscatter fraction has been found to decrease with increasing RH (Fierz-Schmidhauser

et al., 2010a). Fierz-Schmidhauser et al. (2010a) showed that, at Jungfraujoch, the aerosol radiative forcing efficiency (RFE)
increases (in absolute terms) with RH and that this increase is lower if the RH dependence of both the backscattering coefficient
and the single scattering albedo are taken into account in the RFE calculation.

The backscattering enhancement factor ($f_b$(RH, $\lambda$)) is calculated using Eq. 2.

$$f_b(RH,\lambda) = \frac{\sigma_{bsp}(RH,\lambda)}{\sigma_{bsp}(RH_{dry},\lambda)} \tag{2}$$

where $\sigma_{bsp}$ is the hemispheric backscattering coefficient. Again, for simplicity, the backscattering enhancement factor is re-
ferred to as $f_b$(RH) in the text that follows, with the wavelength being 550 nm unless otherwise noted. Most previous studies
investigating the effect of water uptake on the aerosol optical properties focus on the total scattering coefficient, and less atten-
tion is paid to $f_b$(RH). That said, Hegg et al. (1996) noted that, based on Mie theory, $f_b$(RH) would be expected to be lower
than $f$(RH) as it is more sensitive to smaller particles. They were able to see this in a small observational dataset, although

the extent of the difference between $f_b$(RH) and $f$(RH) (40%) was almost double what they expected (25%). Several other
observational studies (Carrico et al., 2003; Koloutsou-Vakakis et al., 2001; Fierz-Schmidhauser et al., 2010a) show that the
backscattering enhancement is significantly lower than the total scattering enhancement across a range of aerosol types. Here,
we present an overview of $f_b$(RH) and $f$(RH) measured across the globe and evaluate them in relation to other collocated
aerosol optical properties.

One important application of experimentally-based $f$(RH) and $f_b$(RH) parameterizations is to improve DARF and visibility
estimates (Kanakidou et al., 2005). Such parameterizations may also be used to evaluate or constrain models (Burgos et al.,
2020) and to predict/estimate the spatial and temporal variability of aerosol hygroscopicity where water uptake measurements
are not available. For example, an $f$(RH) proxy could be used to better adjust vertical profiles of dry aerosol optical properties
to ambient conditions for comparison to remote sensing measurements rather than assuming a constant $f$(RH) throughout the

profile, as is often done when aerosol hygroscopicity measurements are not available (Andrews et al., 2004; Sheridan et al.,
2012). Various approaches have been used to identify proxies for estimating aerosol water uptake, as described below.

Ideally, combined aerosol composition and size distribution measurements together with Mie theory would be used to parame-
terize hygroscopicity, when direct measurements of aerosol hygroscopicity are not available (Zieger et al., 2013). Zieger et al.
(2013) showed that a simple parameterization for all major aerosol types is difficult to retrieve without the knowledge of certain

constraints like the fine mode composition or mode diameter and noted that the coarse mode composition can be an important
parameter in determining the magnitude of the modelled $f$(RH) for total aerosol (Zieger et al., 2013, 2014). Zieger et al. (2010)





demonstrated that, at the Zeppelin station (ZEP, Ny-Alesund), the measured size distribution in conjunction with an assumed chemical composition could also be used as a predictor of $f$(RH).

However, because many measurement sites lack the detailed chemistry, hygroscopic growth and size distribution information
used, for example, in Zieger et al. (2015, 2014) and Fierz-Schmidhauser et al. (2010a, b), other approaches have been used instead. Towards this end, the IMPROVE network developed an equation for hygroscopicity relying on bulk PM2.5 chemical filter measurements of several species, including common ions, crustal elements, black carbon and organic carbon and unspeciated coarse particle mass (Pitchford et al., 2007; Prenni et al., 2019). Quinn et al. (2005) derived a simple parameterization that quantitatively describes the relationship between particulate organic material (POM) mass fraction and $f$(RH) for ambient
aerosols from three different field campaigns when the mass was assumed to consist solely of POM and sulphate. This parameterization was further extended by Zieger et al. (2015) and Zhang et al. (2015b) to include the contribution of additional inorganic components. Those studies demonstrated a decreasing trend of $f$(RH) for increasing POM mass fraction (Quinn et al., 2005; Zhang et al., 2015b; Zieger et al., 2015). Burgos et al. (2020) showed that some global models are unable to reproduce this relationship between $f$(RH) and POM mass fraction (one of the models even simulated the opposite behavior).
While $f$(RH) parameterizations based on chemical and physical properties of the aerosol are useful, high temporal resolution measurements of aerosol composition or hygroscopic growth and complete (fine+coarse) size distribution can be complex and time-consuming to make. They are most frequently collected during field campaigns with duration of a few weeks to a few months. Thus, other potential proxies for $f$(RH), from more widely available observations, are important to investigate. For example, dry aerosol optical properties can provide some qualitative information on particle size and composition and
thus may be useful for constraining $f$(RH). An early example of this was provided by Sheridan et al. (2001) who showed different probability distribution functions of $f$(RH) for different single scattering albedo ($\omega_0$) and sub-micron scattering fraction (strongly correlated with scattering Angström exponent, $\alpha_{sp}$) constraints at Southern Great Plains (SGP), a rural site in the continental US. They linked the decrease in $\omega_0$ to the presence of smoke aerosol and the increase in particle size to the presence of dust aerosol, with both cases showing lower $f$(RH) values than the overall $f$(RH) climatology at the site.
Sheridan et al. (2002) also observed that increases in ($\omega_0$) correlated with increases in $f$(RH) during research flights over the Indian Ocean. Titos et al. (2014b) show a decrease in $f$(RH) together with a decrease in $\omega_0$ at an urban site (Granada, UGR). However, the relationship between $\alpha_{sp}$ and $f$(RH) at UGR varied throughout the year depending on the contribution of dust (coarse) particles at the site (Titos et al., 2014b). Nessler et al. (2005a) reported a strong relationship between $f$(RH) and $\alpha_{sp}$ at Jungfraujoch. During the ACE-Asia campaign, the fine mode fraction (which is highly correlated with $\alpha_{sp}$ (Delene and Ogren,
2002)) was shown to be a good proxy for some hygroscopicity measurements, but less so for others (Doherty et al., 2005). More recently, Titos et al. (2014a) utilized measurements of aerosol optical properties to develop an empirical equation quantifying the relationship between $\omega_0$ and $f$(RH) at a marine site with anthropogenic influence. Titos et al. (2014a) showed, at Cape Cod (PVC, a coastal site in the northeast US), that $f$(RH) increases as the contribution of absorbing particles to aerosol extinction decreases (i.e., as $\omega_0$ increases) and that, at PVC, $\omega_0$ could be used as a proxy to estimate the scattering related hygroscopic
enhancement. In contrast, Zieger et al. (2011, 2014) reported that at Cabauw (CES, in the Netherlands), neither $\omega_0$ nor $\alpha_{sp}$ were good predictors of $f$(RH) (Zieger et al., 2011) while $\alpha_{sp}$ was not a good proxy for $f$(RH) at Melpitz (MPZ, a rural site





in Germany) (Zieger et al., 2014). Zieger et al. (2013) was unable to develop a general parameterization for $f$(RH) based on aerosol optical properties that was valid for multiple sites (Cabauw, Melpitz, Jungfraujoch, Mace Head and Ny-Alesund). In this work, we use the scattering enhancement dataset presented in Burgos et al. (2019) which compiles $f$(RH) and $f_b$(RH)

measurements performed with tandem nephelometer systems at 26 measurement sites. We first describe the variability of scattering enhancement at 23 of these sites in order to present a climatological overview of hygroscopicity observations (3 of the sites of Burgos et al. (2019) dataset are not included in this analysis because of their lower time resolution or because co-located measurements of the aerosol absorption coefficient were not available). This overview includes, for the first time, a climatology of the hygroscopicity of the aerosol hemispheric backscattering coefficient across diverse sites. Additionally, the impact

of measurement wavelength and size cut on scattering related hygroscopicity is assessed. We then combine the hygroscopicity dataset with simultaneous and co-located measurements of dry aerosol optical properties (i.e., dry spectral aerosol light scattering and absorption coefficients) to investigate the impact of relative humidity in radiative forcing calculations. Finally, we extend previous investigations of the viability of using aerosol optical properties (i.e., scattering Ångström exponent and single scattering albedo) as constraints for $f$(RH) across sites and aerosol types.

## 2   Data and methods

We focus on sites from the Burgos et al. (2019) dataset which include $f$(RH) and $f_b$(RH) measurements of $PM_{10}$ (particles with aerodynamic diameter < 10 $\mu$m) or total scattering; some of the measurement sites also have concurrent measurements of $f$(RH) and $f_b$(RH) for $PM_1$ (particles with aerodynamic diameter < 1 $\mu$m). Table 1 lists the sites included in this study and some relevant information for each site. In this section we present a brief overview of the data processing of the $f$(RH) dataset

(more details are provided in Burgos et al. (2019)) and then we describe the dry aerosol optical property dataset.

### 2.1   Global dataset of hygroscopic scattering enhancement

In Burgos et al. (2019), the harmonization of the hygroscopicity datasets was implemented by starting with raw, high resolution wet and dry nephelometer data from the data providers. Data providers shared site log information so that invalid data (e.g., due to instrument failure) could be removed from the dataset. The first processing step was to apply all instrument correc-

tions including truncation and illumination correction (Anderson and Ogren, 1998; Müller et al., 2011), standard temperature and pressure correction and, where applicable, dilution. Wet and dry scattering measurements at low RH were compared to determine an offset for differences in the wet and dry instrument (e.g., due to differential losses in humidifier system). The remaining valid data were fit using an exponential equation (Eq. 3) (Kasten, 1969), where $a$ represents the intercept at RH=0% and $\gamma$ the magnitude of the scattering enhancement. Only humidograms complying with a strict selection criteria were used

for calculating $f$(RH) at RH=85% using Eq. 3. The same procedure was applied to obtain the $f_b$(RH) at RH=85%. The dataset includes scattering and backscattering enhancement factors at three wavelengths (see Burgos et al. (2019) for further details).

$$f(RH, \lambda) = a(1 - RH/100\%)^{-\gamma} \tag{3}$$





The Burgos et al. (2019) dataset includes two $f$(RH=85%) and $f_b$(RH=85%) values: one is referred to a reference RH value of 40% while the other is referred to RH in the range 0-40% (as measured in the reference nephelometer). The difference

between both calculations is small for most of the sites (see Fig. S11 in Burgos et al. (2020)). In this study, we have used the $f$(RH=85%) and $f_b$(RH=85%) referenced to RH 0-40% because of higher data availability and consistency with previous $f$(RH) comparison studies (Titos et al., 2016; Zieger et al., 2013). Finally, Level 2 data as described in Burgos et al. (2019) have been used. Briefly, Level 2 means the data have undergone review, have all corrections applied and are averaged over an appropriate time period for each site (e.g., 1-h averages for sites with high aerosol loading, 6-h averages for very clean sites). In

Burgos et al. (2019), the range in uncertainty (calculated as error propagation) was found to depend on aerosol loading, RH and particle composition, being in the range 25-30% and 25-75% for $f$(RH=85%) and $f_b$(RH=85%), respectively, for moderately hygroscopic aerosol ($\gamma = 0.6$).

## 2.2 Dry aerosol optical property dataset

The sites where the $f$(RH) measurements were performed also typically included measurements of additional aerosol proper-

ties, including dry aerosol scattering and absorption coefficients ($\sigma_{\mathrm{sp}}$ and $\sigma_{\mathrm{ap}}$, respectively). Dry scattering coefficients were needed for the $f$(RH) calculation and were acquired from the data providers as raw data which then underwent the data processing explained in Burgos et al. (2019). Quality-controlled and hourly-averaged absorption data (Level 2) for these sites have been obtained from the EBAS database (www.ebas.nilu.no) or, in the case of the US Department of Energy mobile facility sites (GRW, PVC, PYE, FKB, HLM, HFE, MAO, PGH and NIM), from www.arm.gov/data.

For scattering, all dry nephelometer measurements (except for HYY) were made using TSI integrating nephelometers. The TSI instruments measure total scattering and hemispheric backscattering at three wavelengths (450, 550 and 700 nm). At HYY, an Ecotech Aurora 3000 nephelometer (wavelengths: 450, 525, and 635 nm) was used to obtain dry total scattering. As noted above, the nephelometer data are corrected to account for angular truncation errors and instrument non-idealities using the method proposed by Anderson and Ogren (1998) (for the TSI nephelometers) or by Müller et al. (2011) (for the Ecotech

nephelometer). Table 1 lists the absorption instruments used in this analysis and describes corrections applied in each case. The absorption data corrections are necessary to account for scattering artefacts and other instrument limitations (Bond et al., 1999). For this analysis, both the scattering and absorption data are adjusted to standard temperature and pressure.

From these dry aerosol optical properties, several parameters can be derived which provide qualitative information about inherent characteristics (size and composition) of the aerosol particles: scattering Ångström exponent, backscatter fraction

and single scattering albedo. The equations and a short description of each property is provided below. Sherman et al. (2015) present information on calculating uncertainties for these properties.

The scattering Ångström exponent parameterizes the spectral dependence of light scattering:

$$\alpha_{\mathrm{sp}}(\lambda_1 - \lambda_2) = -\frac{\log \sigma_{\mathrm{sp}}(\lambda_1) - \log \sigma_{\mathrm{sp}}(\lambda_2)}{\log \lambda_1 - \log \lambda_2} \qquad (4)$$

This parameter for atmospheric aerosol particles typically ranges between -1 and 3 and is sensitive to the size distribution of

the aerosol. Values of $\alpha_{\mathrm{sp}}$ near 2 or greater indicate the aerosol is dominated by sub-micrometer particles while $\alpha_{\mathrm{sp}}$ values less





than 1 indicate a significant contribution of coarse mode aerosol to the observed scattering. The scattering Ångström exponent has often been used to differentiate between natural aerosol such as dust or sea salt which tend to dominate the coarse mode and anthropogenic aerosol which consist of smaller (primarily sub-micrometer aerosol) (e.g., Carrico et al., 2003; Anderson and Ogren, 1998). The $\alpha_{sp}$ used here was calculated from the 700 and 450 nm wavelength pair, except for HYY where it was calculated from the 635-450 nm wavelength pair.

The hemispheric backscattering fraction characterizes the amount of light scattered back to the light source:

$$b(\lambda) = \frac{\sigma_{bsp}(\lambda)}{\sigma_{sp}(\lambda)} \tag{5}$$

where $\sigma_{bsp}(\lambda)$ is the hemispheric backscattering coefficient and $\sigma_{sp}(\lambda)$ is the total scattering coefficient at a certain wavelength. Hemispheric backscattered fraction is sensitive to accumulation mode size distribution, especially particles in the 0.1 - 0.4 $\mu m$ size range (Collaud Coen et al., 2007). In this study, $b$ has been calculated for the 550 nm wavelength. Typical values of $b$ for the atmospheric aerosol at this wavelength range from approximately 0.05 to 0.20, with lower values of $b$ indicative of larger accumulation mode particles (i.e., primarily forward scattering) and higher values indicative of smaller accumulation mode particles which backscatter light more efficiently. The hemispheric backscattering coefficient is often used to parameterize the angular distribution of scattered light and has been used to estimate the asymmetry parameter (Andrews et al., 2006).

The single scattering albedo is the ratio of scattering to extinction coefficient (extinction is the sum of scattering and absorption):

$$\omega_0(\lambda) = \frac{\sigma_{sp}(\lambda)}{\sigma_{sp}(\lambda) + \sigma_{ap}(\lambda)} \tag{6}$$

where $\sigma_{ap}(\lambda)$ is the absorption coefficient at wavelength $\lambda$. For atmospheric aerosols, $\omega_0$ typically ranges between 0.5-1.0 (e.g., Laj et al., 2020). Sites dominated by primarily scattering aerosols (e.g., clean maritime sites) exhibit $\omega_0$ values close to 1, with $\omega_0$=1 implying all of the extinction is due to scattering. In contrast, sites impacted by combustion sources have lower $\omega_0$ values. Bond and Bergstrom (2006) suggest that the $\omega_0$ at 550 nm for fresh atmospheric combustion aerosol is in the range of 0.2-0.3. In this study, $\omega_0$ has been calculated for the 550 nm wavelength. The absorption coefficient has been interpolated to this wavelength using the calculated absorption Ångström exponent, $\alpha_{ap}$. For sites performing absorption coefficient measurements at a single wavelength, an $\alpha_{ap}$ of 1 has been assumed. This is a reasonable assumption for anthropogenically-influenced sites where black carbon is the main light absorber, but can differ for sites influenced by dust or biomass burning which show higher spectral dependence (Kirchstetter et al., 2004).

## 2.3 Calculation of RFE RH dependence

In this study, the radiative forcing efficiency, RFE, is calculated following Haywood and Shine (1995):

$$\frac{\Delta F(RH)}{\delta(RH)} \approx -DS_0 T_{atm}^2 (1 - A_C) \omega_0(RH) \beta(RH) \delta(RH) \left\{ (1 - R_S)^2 - \left( \frac{2R_S}{\beta(RH)} \right) \left[ \left( \frac{1}{\omega_0(RH)} \right) - 1 \right] \right\} \tag{7}$$





where the parameters fractional daylight, D, solar flux, $S_0$, atmospheric transmission $T_{atm}$, fractional cloud amount $A_C$, and surface reflectance $R_S$ are independent of the RH. The RH dependent variables are: the aerosol optical depth, $\delta$, upscatter fraction, $\beta$, and single scattering albedo, $\omega_0$.

$\beta$ is calculated from the measured $b$ using the following formula (Sheridan and Ogren, 1999):

$$\beta = 0.0817 + 1.8495b - 2.9682b^2. \tag{8}$$

The radiative forcing efficiency at a certain RH relative to dry conditions (RH<40%) depends on $R_S$, $f$(RH), $\omega_0$(RH), and $b$(RH) in the following way (Sheridan and Ogren, 1999; Fierz-Schmidhauser et al., 2010a):

$$\frac{\frac{\Delta F(RH)}{\delta(RH)}}{\frac{\Delta F(RH<40\%)}{\delta(RH<40\%)}} = \frac{\beta(RH)}{\beta(RH<40\%)} f(RH) \left\{ \frac{(1-R_S)^2 - \left(\frac{2R_S}{\beta(RH)}\right)\left[\left(\frac{1}{\omega_0(RH)}\right)-1\right]}{(1-R_S)^2 - \left(\frac{2R_S}{\beta(RH<40\%)}\right)\left[\left(\frac{1}{\omega_0(RH<40\%)}\right)-1\right]} \right\}. \tag{9}$$

RFE does not take into account that the properties and concentration of aerosol particles vary vertically in the atmospheric column. Fierz-Schmidhauser et al. (2010a) and Luoma et al. (2019) have shown the importance of using the appropriate $R_S$ values at each site. In this study, we have used the average annual value as a function of site type (for rural, urban and mountain sites $R_s = 0.25$, for marine sites $R_s = 0.10$ and for Arctic sites $R_s = 0.65$, based on Hummel and Reck (1979)). These values are just rough estimates and will, of course, vary with the specifics of ground cover at each site, as well as season and factors related to site latitude and altitude.

## 3 Results and Discussion

### 3.1 Overview of $f$(RH) and $f_b$(RH) observations

Figure 1 shows the dry total scattering (top plot) and $f$(RH=85%) statistics for individual sites (bottom plot) obtained over the entire measurement period at each site. Some sites have multiple years of $f$(RH) data, while others performed $f$(RH) measurements just for a few months so the $f$(RH) values shown are not necessarily representative of the annual climatological value at each location. The sites are grouped by their assumed dominant aerosol type (e.g., marine, rural, urban, etc.). Within the groupings, the $f$(RH=85%) values are ordered by the aerosol loading (using the dry aerosol scattering coefficient as a proxy for aerosol amount). The $f$(RH=85%) values shown are for measurements made for total or PM$_{10}$ aerosol.

As we can see in Fig. 1, $f$(RH=85%) values both within and outside of the different site type groupings do not follow a consistent trend with $\sigma_{sp}$ suggesting that aerosol loading does not control the magnitude of $f$(RH=85%). Further, while there are general differences in $f$(RH=85%) as a function of aerosol type, there is also substantial overlap in the $f$(RH=85%) statistics, meaning there is no clear separation in $f$(RH=85%) values amongst most site types. Aerosol particles at Arctic sites tend to have the highest $f$(RH=85%) while urban and a dust-dominated aerosols tend to show the lowest hygroscopicity based on $f$(RH=85%) values. Most clean marine sites (THD, GRW, PYE and MHD) are characterized by higher $f$(RH=85%) than polluted marine sites (KCO and GSN).



Figure 1 is consistent with the general pattern summarized in the Titos et al. (2016) $f$(RH) review paper, in that marine sites

tend to exhibit higher $f$(RH) than rural sites which exhibit higher $f$(RH) than sites dominated by dust laden air masses. The

consistent processing by Burgos et al. (2019) enables presentation of more detailed $f$(RH=85%) statistics than was possible

with literature-reported values provided in the Titos et al. (2016) overview study. However, there is still no clear distinction

of $f$(RH) values as a function of general site types. This is due, at least in part, to variability in aerosol type impacting each

location creating a wide range in observed $f$(RH) at individual sites, as discussed below.

The $f$(RH=85%) values presented in Fig. 1 represent all tandem nephelometer measurements at each site - they have not been

filtered for different aerosol types. This is likely one reason why some relatively clean (based on loading) marine sites (PVC

and CBG) have $f$(RH=85%) values more similar to the polluted marine sites. PVC and CBG are situated relatively close to

each other on the NE coast of North America (Cape Cod and Nova Scotia, respectively). Titos et al. (2014a) show that when

PVC was impacted by anthropogenic emissions from the region, the $f$(RH) values were lower than when the site was affected

by clean marine air. The aerosol at CBG will also vary between clean marine and anthropogenically influenced aerosol as

well as being impacted by biogenic aerosol from forests (Fehsenfeld et al., 2006). While 'clean' vs 'polluted' is a simple

binary way to differentiate air mass types, sites can be impacted by multiple different types of air masses (urban, regional

background, dust, etc.). For example, Zieger et al. (2013) present a table for five European sites detailing $f$(RH) values based

on all measurements at each site and then segregated by the different air mass types identified at each site. Another possible

explanation for the observed variability among sites is that measurements at each site covered different seasons (while some

sites have measurements just for a few months covering one or more seasons, other sites cover more than a year). This is likely

the case at MEL which shows higher $f$(RH=85%) values compared to the other rural sites. The MEL $f$(RH=85%) values are

similar to Arctic (BRW and ZEP) and clean marine sites (e.g., PYE), perhaps because they correspond to winter, when the

aerosol is dominated by inorganic compounds (Zieger et al., 2014).

Also included in Fig. 1 is the median value of the backscattering enhancement factor, $f_b$(RH=85%), indicated by the star over-

laid on the $f$(RH=85%) plots. Box-whisker plots of $f_b$(RH) are available in the supplementary material, Fig. S1. The backscat-

tering enhancement factor is useful as it indicates how the angular distribution of aerosol light scattering changes with RH and

is thus a key factor in aerosol forcing calculations for ambient atmospheric conditions. In general, the $f_b$(RH=85%) values

track the $f$(RH=85%) variations at each site, with the median $f_b$(RH=85%) always being lower than the median $f$(RH=85%).

It should be noted that the $f$(RH=85%) and $f_b$(RH=85%) information in Fig. 1 refers to all available $f$(RH=85%) and all

$f_b$(RH=85%) values at each site and the two parameters do not necessarily have the same temporal coverage - there were

fewer successful fits of $f_b$(RH=85%) than there were of $f$(RH=85%) (Burgos et al., 2019). The percentage of $f_b$(RH=85%)

coinciding with $f$(RH=85%) ranges from <10 % at JFJ and ZEP to >75 % at CES and MEL. The $f_b$(RH=85%) measurements

at HYY and MAO were not included due to instrument issues.

The relationship between $f$(RH=85%) and $f_b$(RH=85%) is further explored in Fig. 2. This figure shows that there is a linear

trend between $f$(RH=85%) and $f_b$(RH=85%) with a slope of 0.58 $\pm$ 0.12, intercept of 0.31 $\pm$ 0.23 and strong correlation

($R^2 = 0.81$). These parameters have been retrieved from a weighted bivariate fit according to York et al. (2004), taking the

standard deviation of the average values as an input for the uncertainty calculation. Note that for this analysis, only data with co-



incident $f$(RH=85%) and $f_b$(RH=85%) measurements are included. The relationship between $f$(RH=85%) and $f_b$(RH=85%)

is supported by observations from other sites reported in the literature and these have been added to Fig. 2 (grey dots). Similar relationships between $f_b$(RH=85%) and $f$(RH=85%) are also observed for the temporally matched data points for the individual sites (Fig. S2 of the supplementary material shows some examples). Hegg et al. (1996) suggest that Mie theory predicts a reduction of approximately 25 % in $f_b$(RH) relative to $f$(RH) for typical atmospheric aerosols, but did not have access to a data base of aerosol water uptake impact on aerosol optical properties such as Burgos et al. (2019) to demonstrate the relationship.

Figure 2 exhibits a general pattern relating $f$(RH) and $f_b$(RH), which might be useful due to the scarcity of $f_b$(RH) measurements. Hegg et al. (1996) noted that enhanced reductions in $f_b$(RH) could confound the attribution of aerosol water content in aerosol optical depth (AOD) retrievals from backscattered radiation if water uptake assumptions were based on $f$(RH) rather than $f_b$(RH). At that time, Hegg et al. (1996) also noted that many models implicitly assumed that the humidity dependence of backscattering was identical to total scattering. Later, Wang and Martin (2007) suggested that satellite retrievals make as-

sumptions about RH when they use algorithms to retrieve aerosol information from measured reflectances. If the aerosol is at a different RH then what the satellite algorithms assume for the aerosol properties (like backscattering) may not be appropriate and could lead to incorrect retrievals. Wang and Martin (2007) noted that in satellite retrieval algorithms for AOD that employ angular-dependent radiance observations, the aerosol hygroscopicity must be explicitly considered.

### 3.2 Impact of size cut and wavelength on $f$(RH)

The value of $f$(RH) is controlled by both the size and composition of the aerosol particles, although size and composition are not completely independent variables. Further, Mie theory dictates that, for a given composition and concentration, aerosol scattering will depend on both the size of the particle and the wavelength of the incident light. Thus, there may be some information about the character of the underlying aerosol gained by studying differences in $f$(RH) as a function of measurement size cut and spectral dependence.

The size split between $PM_1$ and $PM_{10}$ is roughly a size split between anthropogenic and natural aerosol (e.g., Carrico et al., 2003; Anderson and Ogren, 1998), although at clean marine sites this is less true (due to secondary particle formation from natural emissions and sub-$\mu m$ sea salt). Thus, understanding the difference in hygroscopicity between these two size cuts may provide information about the different behaviors of man-made and natural particles in a humid atmosphere. Figure 3 shows the difference between $f$(RH=85%)$_{PM_1}$ and $f$(RH=85%)$_{PM_{10}}$ at 550 nm as a function of dry $\alpha_{sp}$, binned in 0.2 $\alpha_{sp}$ increments.

As in Andrews et al. (2011), only bins that have a standard error less than a certain threshold (3% in this case) of the typical value of that variable were included (standard error is the standard deviation of the sample divided by the square root of the number of points in the sample). For simplicity, a typical value of $\alpha_{sp}$ was assumed to be 2.0 meaning 3 % of the typical value is 0.06. Bins with a larger standard error were omitted, since they may not be representative of actual aerosol systematic variability at the site.

Across all sites the difference in $f$(RH=85%) between $PM_1$ and $PM_{10}$ tends to be positive indicating that $PM_1$ particles exhibit relatively more scattering enhancement than their $PM_{10}$ counterparts. This is consistent with previous $f$(RH) values reported in the literature where both $PM_1$ and $PM_{10}$ $f$(RH) were measured (Carrico et al., 2000; Koloutsou-Vakakis et al., 2001; Carrico





et al., 2003; Titos et al., 2014a; Jefferson et al., 2017) and is consistent with Mie-modelling of $f$(RH) (Zieger et al., 2010, 2013). This may seem counter-intuitive (for example, at marine sites where the $PM_{10}$ aerosol will be dominated by very

hygroscopic coarse mode sea salt). However, it is consistent with Mie theory where a stronger increase in scattering efficiency in the accumulation mode size range would be expected while the scattering efficiency for super-micrometer aerosol at visible wavelengths is relatively constant. This results in smaller particles with smaller diameter change due to water uptake exhibiting higher scattering enhancement due to a larger increase in scattering efficiency than bigger particles with more diameter growth but with almost no change in the scattering efficiency. Zieger et al. (2013) show this effect by plotting $f$(RH) as a function

of particle size for some common atmospheric constituents (their Figure 2 and Table 3). The compensating effects of size and hygroscopicity have been observed and explained for Arctic aerosol (Zieger et al., 2010), where smaller but less hygroscopic particles exhibit a similar $f$(RH) as larger, more hygroscopic particles.

Figure 3 also shows that, at marine and Arctic sites, the separation between $PM_1$ and $PM_{10}$ $f$(RH=85%) is more variable and exhibits a dependence on $\alpha_{sp}$. The $f$(RH=85%) $PM_1$-$PM_{10}$ separation increases as $\alpha_{sp}$ decreases, which likely suggests that

the marine sea salt aerosol is not confined to the coarse mode or there are other hygroscopic components in the fine marine aerosol. The behavior observed is also consistent with Carrico et al. (2000, 2003) and Titos et al. (2014a) reporting larger differences between $PM_1$ and $PM_{10}$ $f$(RH) for clean marine air (lower $\alpha_{sp}$) than for polluted marine air (higher $\alpha_{sp}$). Quinn et al. (2002) showed that at BRW that, at some times of year, the sea salt contributes as much to the $PM_1$ aerosol as it does to the super-$\mu m$ aerosol.

Differences in $f$(RH=85%) for the two size cuts as a function of $\alpha_{sp}$ are not observed for rural and urban sites (Fig. 3). The likely explanation for this is twofold. First, the $f$(RH=85%)$_{PM_1}$ at these sites is probably dominated by organic and black carbon aerosols which are typically less hygroscopic than soluble components such as sulfate (Zieger et al., 2013). Second, when significant coarse aerosol is present at these sites, it most likely is dominated by dust which is also less hygroscopic (Titos et al., 2014b; Fierz-Schmidhauser et al., 2010a). Both these factors would result in little observed difference between $PM_1$ and

$PM_{10}$ hygroscopicity at sites with these characteristics. This is consistent with Koloutsou-Vakakis et al. (2001) finding minimal difference between $PM_1$ and $PM_{10}$ $f$(RH) at a rural site in central Illinois (USA).

Understanding spectral changes in $f$(RH) is also important since surface solar irradiance and, hence, radiative forcing is a function of wavelength (e.g., Kiehl and Briegleb, 1993; Kotchenruther and Hobbs, 1998). Some previous reports on the spectral dependence of $f$(RH) describe diversity in spectral dependence for different air mass types (e.g., Carrico et al., 2003, 2000;

Fierz-Schmidhauser et al., 2010a; Magi and Hobbs, 2003), but generally, most studies do not discuss their spectral findings in detail or put them in a wider context. Our review of the literature indicates that in the majority of cases where spectral values of $f$(RH) are presented, $f$(RH) increases for increasing wavelength (Carrico et al., 1998, 2000; Kotchenruther et al., 1999; Koloutsou-Vakakis et al., 2001; Zieger et al., 2014, 2015; Magi and Hobbs, 2003). There are a few cases in the literature where the spectral dependence of $f$(RH) is found to be negligible: polluted marine (Carrico et al., 2003), smoke (Kotchenruther

and Hobbs, 1998), and clean Arctic (Zieger et al., 2010). There are even fewer cases where the spectral dependence of $f$(RH) increases with decreasing wavelength: two occurred during dust impacted $f$(RH) measurement periods at different sites (in Asia (Carrico et al., 2003) and at a high alpine site (JFJ) (Fierz-Schmidhauser et al., 2010a)). Fierz-Schmidhauser et al. (2010a) also





reported this type of spectral dependence at JFJ for some times that were not impacted by dust; they attributed this to shifts in aerosol size distribution, but did not associate it with a specific aerosol type.

Figure 4 provides an overview of spectral dependencies across the range of sites and aerosol types studied here by presenting the frequency of occurrence of the difference in spectral $f$(RH=85%) at 700 nm with the $f$(RH=85%) at 450 nm. These wavelengths bracket the 550 nm data presented in other plots and represent the extreme of the wavelength dependence available with this dataset. Figure 4 shows that the $f$(RH=85%) at 700 nm is typically larger than the $f$(RH=85%) at 450 nm for most sites. Similar behavior is observed for $f_b$(RH=85%) (not shown here). For the marine sites, Fig. 4 suggests that the wavelength

difference of $f$(RH=85%) is always positive except for GSN - a site which was likely impacted by dust during the measurements (e.g., Doherty et al., 2005). Rural and urban sites show a higher variability in the observed wavelength dependence, with most sites showing datapoints for which the difference $f$(RH=85%, 700 nm) - $f$(RH=85%, 450 nm) is negative (frequency of occurrences are shifted towards more negative values relative to what is observed for marine sites, but on average the differences are positive). Sites like UGR (urban) and SGP (rural) show frequency distributions centred around 0. This could be due

to the influence of dust particles at these sites (Sheridan et al., 2001; Titos et al., 2014b). For JFJ, we found that, for most of the measurements, the $f$(RH=85%, 700 nm) is larger than $f$(RH=85%, 450 nm) in agreement with Bukowiecki et al. (2016), although measurements showing the opposite behavior also exist. This could be associated with dust-influenced periods as shown by Fierz-Schmidhauser et al. (2010a). It should be noted that the measurement period at JFJ in Fierz-Schmidhauser et al. (2010a) is different from the data time frame used in this study. In general, the wavelength dependence observed for

$f$(RH=85%) across sites is constrained within a $f$(RH=85%) range of +/- 0.5, but the wavelength dependence is smaller for most sites that show narrower frequency distributions (exceptions are GSN, UGR and SGP).

### 3.3 Changes in RFE relevant properties as a function of RH

Figure 5 shows the RH dependence of the single scattering albedo, backscatter fraction, and the ratio of the radiative forcing efficiency at a certain RH to dry conditions (RH<40%) for different site types. Absorption is assumed to be independent of

RH and the potential absorption enhancement due to a water coating is neglected. This is a simplified assumption since the absorption enhancement due to coating and water uptake can be a very complex process and it strongly depends on the coating material and the RH history at which the absorbing particle has been exposed (Yuan et al., 2020). Nevertheless, several authors (e.g., Nessler et al., 2005b; Yuan et al., 2020) have reported that the absorption enhancement is minimal compared to the scattering enhancement, and calculations of RFE usually assume that the RH absorption enhancement can be neglected (e.g.,

Fierz-Schmidhauser et al., 2010a; Luoma et al., 2019). Yuan et al. (2020) reported that the RH scattering enhancement of black carbon coated with ammonium nitrate was 5-fold the enhancement observed in the absorption coefficient. Since measurements of absorption RH enhancement are not available for this study, we assumed that the absorption was not dependent on RH as done in previous studies and the results have been consequently discussed on the light of this simplification.

Overall, as shown in Fig. 5, $\omega_0$ increases with RH for all site types, with a larger slope for some marine (KCO and GSN),

urban (UGR), and rural sites (CES, HLM). The increase is, of course, because of the enhanced scattering due to water uptake simultaneous with the assumed lack of change in aerosol absorption. Any enhancement in the absorption coefficient due to





water uptake will result in a lower $\omega_0$ enhancement with RH than observed in Fig. 5. In contrast, $b$ shows opposite behavior to $\omega_0$ (i.e., $b$ decreases with increasing RH) because as particles grow the amount of light scattered in the backward direction is reduced. Some sites show less dependence of $b$ on RH than others but there is no consistent pattern with site type, suggesting a

complex interplay of aerosol composition and size distribution. Similar RH dependence of $\omega_0$ and $b$ has been shown in previous studies at individual sites (e.g., Carrico et al., 2003; Fierz-Schmidhauser et al., 2010b, a).

All sites show an increase in the forcing efficiency at elevated RH, indicating the importance of considering the RH effect in the aerosol optical properties when estimating aerosol forcing. The forcing efficiency is negative for all sites, indicative of a cooling effect which becomes larger (in absolute terms) with the increase of RH. However, there does not appear to be a

clear trend related to site type. The range of forcing enhancement among sites varies from almost no enhancement up to a factor of 3-4 at RH=90 %. The enhancement in forcing efficiency due to RH at each site is modulated by the RH dependence of $\omega_0$ and $b$. As noted by Luoma et al. (2019), the tendencies of $\omega_0$ to increase with RH and of $b$ to decrease with RH will have opposite effects on the aerosol radiative forcing efficiency and, thus, to some extent the RH dependencies of these two parameters will counterbalance each other. However, as shown in Fig. 5, the differences in the RH dependence of $\omega_0$ and $b$

can result in significant changes in the forcing efficiency. For example, CES and APP exhibit very similar trends of $b$ with RH, but CES exhibits a stronger increase of $\omega_0$ with RH than is observed at APP. This difference results in a larger increase in the forcing efficiency due to RH at CES, the site which shows the largest enhancement in the forcing efficiency at high RH (4-fold) relative to dry conditions. Sites with higher $f$(RH) values do not necessarily exhibit higher forcing efficiency enhancement at elevated RH. For example, ZEP and JFJ show similar forcing efficiency enhancement but the $f$(RH) values

at ZEP are much higher than those at JFJ (see Fig. 1). Figure S3 of the supplementary material shows the forcing efficiency enhancement as a function of $f$(RH). This figure demonstrates that the forcing efficiency dependence on $f$(RH) is different from site to site. Rural and urban sites show a higher rate of increase of the forcing enhancement per $f$(RH) (curves above the 1:1 line) while for marine and Arctic sites most of the curves lie below the 1:1 line (exceptions are BRW and PVC). In other words, for the same $f$(RH) value the impact in the radiative forcing would be higher at rural/urban sites than at marine/Arctic

sites. This is likely due to the lower $\omega_0$ at rural and urban sites that show a significant increase with RH, while marine and Arctic sites are characterized by $\omega_0$ values already close to 1, so the effect of RH is lower. Luoma et al. (2019) shows that at HYY the seasonal variability of parameters impacting RFE (aerosol and location specific parameters such as cloud cover and sun angle) can have a profound impact on the resulting radiative forcing. Since we have kept the focus on the effect of RH on the radiative forcing efficiency, assumptions regarding those parameters were not necessary (see Equation 9). Nevertheless, the

simplified assumptions regarding $R_S$ values which we know depend on season, surface properties, altitude and latitude, as well as neglecting the potential RH dependence of the absorption coefficient, might have an impact on the results obtained. The latter assumption is expected to have a small effect in the calculated RFE because the scattering is dominant in the calculation of $\omega_0$ and it is expected to show higher RH dependence than the absorption coefficient. Concerning the use of a constant $R_S$ value for each site type, Fierz-Schmidhauser et al. (2010a) showed that at JFJ, changes in $R_S$ between 0.05 and 0.25 lead to changes

in the RFE between 2.2 and 2.5, approximately. Therefore, the results obtained in this study of the RFE at multiple sites around the world offer a comprehensive picture of the importance of RH on RFE, despite the intrinsic limitations discussed above.





### 3.4 Relationship between $f(\text{RH=85\%})$ and dry aerosol optical properties

Predicting aerosol hygroscopicity is critical for understanding aerosol/water interactions which impact climate, visibility and cloud formation. As shown in Burgos et al. (2020), there is a large diversity in $f(\text{RH=85\%})$ simulated by Earth system models.

The diversity among different models is primarily driven by differences in model hygroscopicity parameterizations and model chemistry (Burgos et al., 2020). Aerosol hygroscopicity measurements (such as the Burgos et al. (2019) dataset used here) are not very common, so developing methods to estimate hygroscopicity from other measurements would be helpful in setting constraints for models and explaining physical processes in the atmosphere. Our approach here is to utilize aerosol optical measurements such as $\omega_0$ and $\alpha_{\text{sp}}$ which indirectly provide qualitative information about aerosol composition and size. As

noted in the introduction, there has been some success with this approach for individual stations (e.g., Sheridan et al., 2002, 2001; Titos et al., 2014a; Doherty et al., 2005; Nessler et al., 2005b). However, there have also been some efforts where this approach did not work for individual sites (Zieger et al., 2014, 2011; Doherty et al., 2005) or across several sites (Zieger et al., 2013). Here, with a much larger and more diverse group of sites, we further explore relationships between $f(\text{RH=85\%})$ and several, readily available, observed aerosol optical properties.

Figure 6 shows $f(\text{RH=85\%})$ segregated by $\omega_0$ (upper panel) and by $\alpha_{\text{sp}}$ (lower panel). The segregation has been performed based on the 25th and 75th percentiles of $\omega_0$ and $\alpha_{\text{sp}}$, respectively, at each site, assuring enough data in each category while looking at relatively extreme situations. The values of the 25th and 75th percentiles for each site are noted in Fig. 6. The t-test (at 5% significance level) has been used here to determine if the segregation of $f(\text{RH=85\%})$ either by $\omega_0$ or $\alpha_{\text{sp}}$ is statistically significant (i.e., if the $f(\text{RH=85\%})$ values for the <25th percentile category and the $f(\text{RH=85\%})$ values for the >75th percentile

category are statistically different from each other). We also tested if the $\omega_0$ and $\alpha_{\text{sp}}$ values were statistically different. According to the results of the t-test, all $\omega_0$ and $\alpha_{\text{sp}}$ divisions are statistically significant while $f(\text{RH=85\%})$ segregation was not statistically different for ZEP and PGH for $\omega_0$ separation and for GSN, MEL, CES, HFE and MAO for $\alpha_{\text{sp}}$ separation. The significance tests failed at NIM because of limited data. The sites showing non-statistically significant differences are represented with thinner lines in Fig. 6.

At the majority of sites (75% of the sites) higher hygroscopicity, based on the median value of $f(\text{RH=85\%})$, is observed for aerosols with higher values of $\omega_0$. This is consistent with the idea that scattering aerosol are often composed of soluble ions (e.g., sea salt and sulphates) which are hygroscopic while combustion aerosols which contribute to lower $\omega_0$ tend to be less to non-hygroscopic. However, the opposite trend is observed (i.e., higher $f(\text{RH=85\%})$ for lower $\omega_0$ values) at five sites (JFJ, CBG, KCO, MAO and HYY). Although for CBG, KCO and MAO the difference between the $f(\text{RH=85\%})$ segments is small,

it is statistically significant. These five sites are quite different in terms of site type (high altitude, marine, rural) making it difficult to identify an obvious reason for their different behavior with respect to $\omega_0$. Additionally, as noted above, general site categories can encompass a variety of air mass types depending on transport and seasonality, so further investigation (e.g., trajectory analysis and aerosol chemistry information similar to Zieger et al. (2013)) would be needed. That level of detail is beyond the scope of this work.





Figure 6 also shows $f$(RH=85%) segregated by $\alpha_{sp}$. In general, Fig. 6 shows that approximately half of the sites exhibit larger $f$(RH=85%) values when the $\alpha_{sp}$ is lower (this is the case for most marine sites). Lower $\alpha_{sp}$ values indicate the presence of coarse aerosol. Depending on the site, coarse aerosol could be associated with sea salt which is hygroscopic and would lead to higher $f$(RH). However, lower $\alpha_{sp}$ may also be associated with the presence of dust aerosol which is often considered to be non-hygroscopic depending on age and atmospheric processing (Fierz-Schmidhauser et al., 2010a; Titos et al., 2014b).

The range in hygroscopicity properties for different types of coarse particles explains why $\alpha_{sp}$ is less useful as a constraint of $f$(RH=85%) than $\omega_0$ (Zieger et al., 2013, 2014).

Figure 7 shows the scatter plot of the median $f$(RH=85%) versus $\omega_0$ color coded by $\alpha_{sp}$ for the sites considered in this study. At PVC, Titos et al. (2014a) found that aerosol scattering related hygroscopicity followed an exponential relationship with $\omega_0$, and that particles with higher $f$(RH) and $\omega_0$ also tended to have the lowest $\alpha_{sp}$ values (predominance of bigger particles). When

taking into account all the median values for all sites of this study, Fig. 7 shows a similar pattern, where more hygroscopic and less absorbing particles tend to be larger (lower $\alpha_{sp}$). This is true for most marine sites, while Arctic sites (BRW and ZEP) show a similar behavior but exhibit lower $\omega_0$ values than the marine sites. Other sites, characterised by considerably lower $\omega_0$ than PVC, such as UGR, HLM, FKB or KCO exhibit higher scattering enhancement than predicted by the exponential relationship observed at PVC (Titos et al., 2014a). Additional exceptions to this trend are CES and MEL, which have much

higher scattering enhancement than would be estimated based on their respective $\omega_0$ or $\alpha_{sp}$ values. It is important to bear in mind that using the median value for each site hides the strong variability in aerosol properties at each site (as shown by the large standard deviations shown in Fig. 7) and that the separation of the individual sites' data sets by wind direction or air mass origin may improve the pattern observed. The individual relationship between $f$(RH=85%) and $\omega_0$ for each site is shown in Fig. S4. Most sites show no trend between the analysed variables, except for some of the marine sites (e.g., PVC, PYE, GRW,

MHD and THD). Although using aerosol optical properties as an overall predictor of $f$(RH=85%) seems of limited utility based on the high variability observed among sites, the patterns we observe between $f$(RH=85%) and $\alpha_{sp}$ and especially $\omega_0$ may still be a useful constraint models on a site by site basis.

## 4 Conclusions

In this paper we have presented an extended overview and analysis of the range and variability of the scattering enhancement

factor ($f$(RH=85%)) at 23 diverse sites across the globe based on the data set developed by Burgos et al. (2019). There are not clear patterns in $f$(RH=85%) as a function of site type, although in general marine sites tend to exhibit higher $f$(RH=85%) than rural and urban sites. The variability in $f$(RH=85%) observed at each site suggests that simple assumptions about $f$(RH) based on dominant aerosol type will not capture the actual range observed in this parameter for a given location.

We have also studied in detail the climatology of the hemispheric backscattering enhancement factor $f_b$(RH=85%) across the

same set of diverse sites. The value of $f_b$(RH=85%) is highly correlated with that of $f$(RH=85%), and the difference between $f_b$(RH=85%) and $f$(RH=85%) increases with $f$(RH=85%) suggesting that the more hygroscopic the aerosol, the lower the





scattering enhancement in the backward direction relative to the total scattering enhancement, which is in agreement with Mie theory and consistent with previous observations at individual sites.

We investigated the influence of size cut and wavelength on $f$(RH=85%). $f$(RH=85%) is found to be higher for PM$_1$ than for PM$_{10}$ at most sites, which is a result of the size dependence of scattering efficiency according to Mie theory. Specifically at marine and Arctic sites a larger difference in $f$(RH=85%) as a function of size cut is observed for lower $\alpha_{sp}$ (predominance of larger particles), which is explained by the fact that sea salt is not only confined to the coarse mode. Small differences in $f$(RH=85%) for rural and urban sites are observed as a function of size cut. Additionally, $f$(RH=85%) increases as wavelength increases for the pair of wavelengths we studied here (450 and 700 nm). In contrast, the spectral dependence is negligible or even shows the opposite pattern for sites impacted by dust.

The light scattering enhancement influences the estimates of direct radiative aerosol forcing by changing the single scattering albedo and the angular distribution of scattered light. We assessed this influence across the sites in this study and found that $\omega_0$ increases with increasing RH while $b$ decreases. These results are in agreement with previous studies performed at individual sites. The aerosol radiative forcing efficiency (RFE) at high RH is larger (more negative) than the dry RFE, although the magnitude of the RH effect can be rather small in some cases, while for other cases a factor of 3-4 enhancement in RFE is observed (e.g., for clean Arctic sites). This RFE enhancement with RH is more pronounced at sites with lower $\omega_0$ values despite the lower $f$(RH) values at those sites. In spite of the simplified assumptions made to calculate RFE (constant $R_s$ value and neglecting any absorption enhancement due to water uptake) we present an analysis of the effect of RH on RFE at very diverse sites, demonstrating the importance of considering RH in the calculation of RFE, especially at sites with low $\omega_0$.

We explored the relationship between readily available in situ aerosol optical properties (single scattering albedo ($\omega_0$) and scattering Ångström exponent ($\alpha_{sp}$)) and $f$(RH=85%). The $f$(RH=85%) values for each site were segregated by the outer quartile ranges of co-located $\omega_0$ and $\alpha_{sp}$ properties. With a few exceptions, lower $\omega_0$ tended to be associated with lower $f$(RH=85%) consistent with the idea that combustion-related aerosol tends to be less hygroscopic. Splitting $f$(RH=85%) by $\alpha_{sp}$ was less definitive. At marine-influenced sites, lower $\alpha_{sp}$ (likely indicating the presence of sea salt) tended to coincide with higher $f$(RH=85%). At other sites which may have a dust influence the opposite was observed, i.e., lower $\alpha_{sp}$ was associated with lower $f$(RH=85%). Further information on aerosol size, chemical composition or air mass history would help to better constrain the relationship between $f$(RH=85%) and other variables.

This study provides a detailed analysis of multi-wavelength aerosol scattering and backscattering enhancement as a function of RH based on the harmonized dataset devoleped in Burgos et al. (2019). This dependence on RH is important for radiative forcing estimations. Measurements of $f$(RH) are necessary to address the impact of RH on aerosol optical properties. The relationship between $f$(RH=85%) and other aerosol optical properties such as $\omega_0$ or $\alpha_{sp}$ could be useful to constrain $f$(RH) values within Earth system global models, but appear to be of limited utility for predictions of $f$(RH) in the absence of direct $f$(RH) measurements. Future studies with this $f$(RH) dataset could explore other aerosol properties like size distribution (fine + coarse) and aerosol chemistry as proxies of $f$(RH). Furthermore, separation of each dataset by predominant wind direction or air-mass history could help to better constrain the relationship between $f$(RH) and other aerosol properties.



*Code and data availability.* The hygroscopicity dataset used in this study is already publicly available (see Burgos et al., 2019). Absorption data was obtained from ebas database (http://ebas.nilu.no) or DOE archive (www.arm.gov/data).

*Author contributions.* G.T., M.B. and E.A. performed the data analysis. G.T., E.A., M.B., and P.Z. designed study and wrote the paper. E.A., L.A.A., U.B., B.H., A.J., J.S., G.T., E.W., K.L., C.O and A.W. were involved in either the humidified nephelometer or the absorption coefficients measurements (or both). All authors read and commented on the manuscript.

*Competing interests.* The authors declare that they have no conflict of interest.

*Acknowledgements.* This work was essentially supported by the Department of Energy (USA) under the project DE-SC0016541.



## 5   Tables

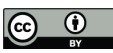

**Table 1.** Table listing sites, site type, references (general reference and $f$(RH) reference if available), absorption instruments used at each site and correction, and cut size. Site order by site type and within it, alphabetically. PSAP=Particle Soot Absorption Photometer (1-W and 3-W refers to single and 3 wavelengths versions of the instrument), MAAP=Multi-Angle Absorption Photometer. B1999= Bond et al. (1999), O2010= Ogren (2010), SS2007=Springston and Sedlacek (2007). Instruments change date [1]October 2005, [2]March 2016, [3]April 2005

| Station ID | Station Name, Country | Site type | Reference | Absorption Instrument | Correction | Cut Size |
|---|---|---|---|---|---|---|
| BRW | North Slope of Alaska, USA | Arctic | Schmeisser et al. (2018) | 3-W PSAP | B1999,O2010 | $PM_{10}$, $PM_1$ |
| ZEP | Zeppelin, Norway | Arctic | Schmeisser et al. (2018); Zieger et al. (2010) | 1-W PSAP | B1999,O2010 | None |
| JFJ | Jungfraujoch, Switzerland | Mountain | Fierz-Schmidhauser et al. (2010a); Bukowiecki et al. (2016); Zieger et al. (2012) | MAAP | n/a | None |
| CBG | Chebogue Point, Canada | Marine | Ervens et al. (2010); Fehsenfeld et al. (2006) | 1-W PSAP | B1999,O2010 | $PM_{10}$, $PM_1$ |
| GRW | Graciosa, Portugal | Marine | Wood et al. (2015) | 3-W PSAP | B1999,O2010 | $PM_{10}$, $PM_1$ |
| GSN | Gosan, S. Korea | Marine | Doherty et al. (2005) | 1-W PSAP | B1999,O2010 | $PM_{10}$, $PM_1$ |
| KCO | Kaashidhoo Climate Obs., R. Maldives | Marine | Clarke et al. (2002), Eldering et al. (2002) | 1-W PSAP | B1999,O2010 | $PM_{10}$, $PM_1$ |
| MHD | Mace Head, Ireland | Marine | Jennings et al. (2003); Fierz-Schmidhauser et al. (2010b) | MAAP | n/a | None |
| PVC | Cape Cod, USA | Marine | Titos et al. (2014a) | 3-W PSAP | B1999,O2010 | $PM_{10}$, $PM_1$ |
| PYE | Point Reyes, USA | Marine | Berkowitz et al. (2011) | 3-W PSAP | B1999,O2010 | $PM_{10}$, $PM_1$ |
| THD | Trinidad Head, USA | Marine | Parrish et al. (2004) | 1-W and 3-W PSAP[1] | B1999,O2010 | $PM_{10}$, $PM_1$ |
| APP | Appalachian State, USA | Rural | Sherman et al. (2015) | 3-W PSAP and CLAP[2] | B1999,O2010 | $PM_{10}$, $PM_1$ |
| CES | Cabauw, Netherlands | Rural | Pandolfi et al. (2018); Zanatta et al. (2016); Zieger et al. (2011) | MAAP | n/a | $PM_{10}$ |
| FKB | Black Forest, Germany | Rural | Fierz-Schmidhauser et al. (2010) | 3-W PSAP | B1999,O2010 | $PM_{10}$, $PM_1$ |
| HLM | Holme Moss, UK | Rural | Liu et al. (2011) | 3-W PSAP | B1999,O2010 | $PM_{10}$, $PM_1$ |
| HYY | Hyytiälä, Finland | Rural | Luoma et al. (2019); Zieger et al. (2015) | 3-W PSAP | B1999,SS2007 | None |
| LAN | Lin'an, China | Rural | Zhang et al. (2015a) | n/a | n/a | $PM_{10}$ |
| MEL | Melpitz, Germany | Rural | Pandolfi et al. (2018); Zanatta et al. (2016); Zieger et al. (2014) | MAAP | n/a | $PM_{10}$ |
| SGP | Southern Great Plains, USA | Rural | Sherman et al. (2015); Jefferson et al. (2017) | 1-W and 3-W PSAP[3] | B1999,O2010 | $PM_{10}$, $PM_1$ |
| HFE | Shouxian, China | Urban | Liu and Li (2018) | 3-W PSAP | B1999,O2010 | $PM_{10}$, $PM_1$ |
| MAO | Manacapuru, Brazil | Urban | Almeida et al. (2019) | 3-W PSAP | B1999,O2010 | $PM_{10}$, $PM_1$ |
| PGH | Nainital, India | Urban | Dumka et al. (2017) | 3-W PSAP | B1999,O2010 | $PM_{10}$, $PM_1$ |
| UGR | Granada, Spain | Urban | Titos et al. (2012, 2014b) | MAAP | n/a | None |
| NIM | Niamey, Niger | Desert | Miller and Slingo (2007) | 3-W PSAP | B1999,O2010 | $PM_{10}$, $PM_1$ |



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





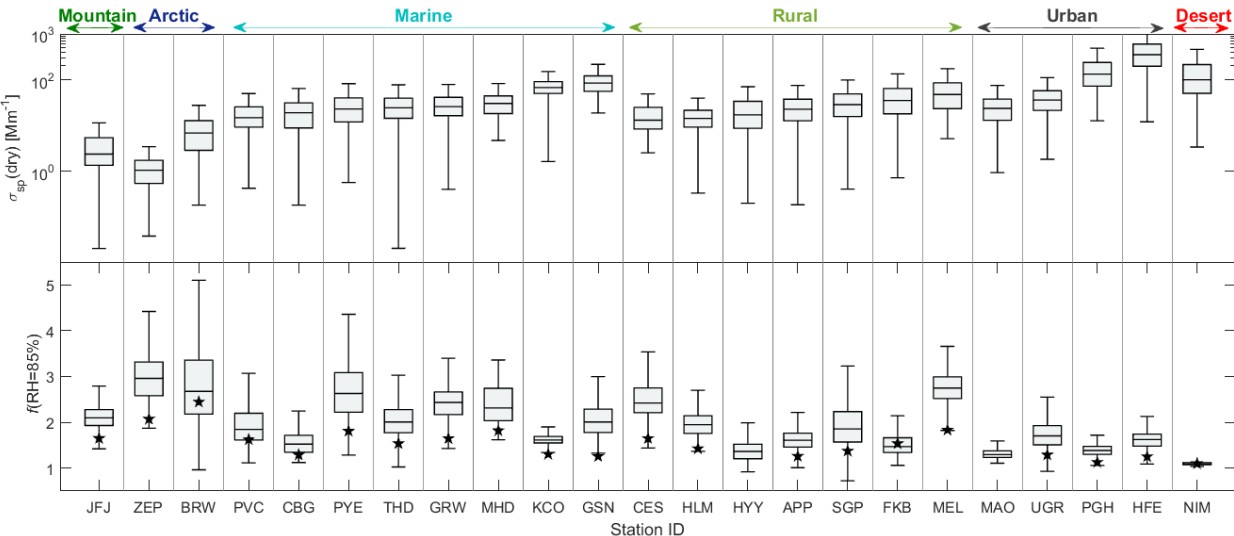

**Figure 1.** Boxplot of dry scattering coefficient (upper panel) and $f$(RH=85%) (lower panel) at $\lambda$ =550 nm ($\lambda$ =525 nm at HYY) . The black stars in the lower panel indicate the median backscattering enhancement factor $f_b$(RH=85%). Sites are sorted by site type and scattering coefficient (from low to high). For each box, the central mark is the median, the box extends vertically between the 25th and 75th percentiles, the whiskers extend to the most extreme data that are not considered outliers.

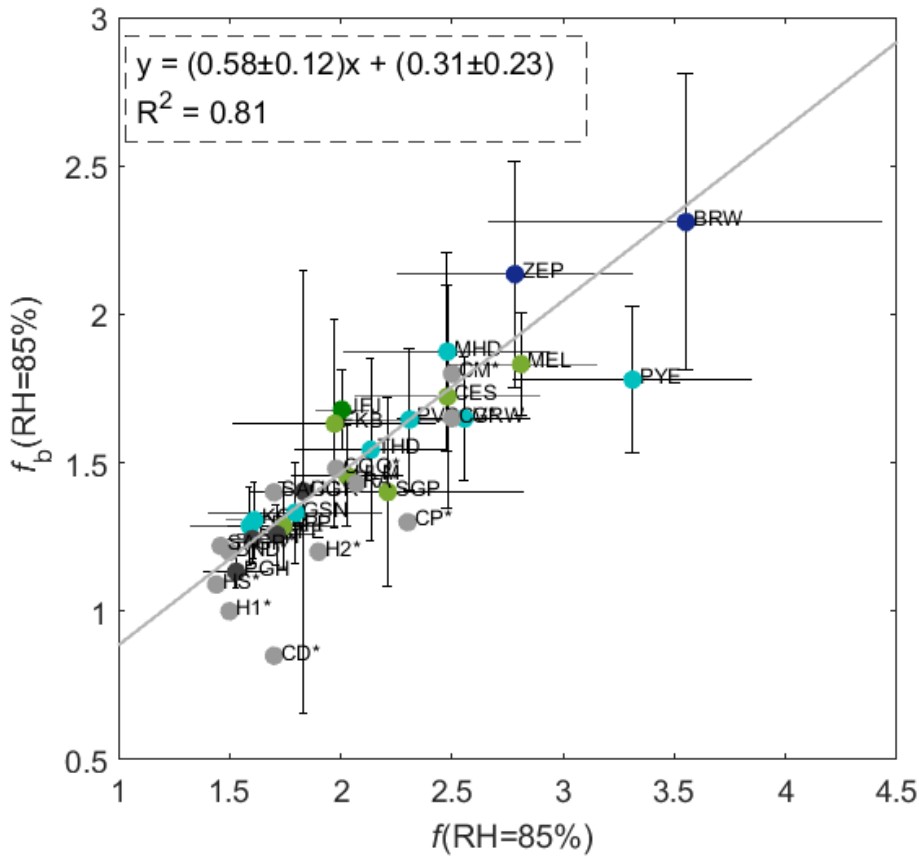

**Figure 2.** Scatterplot of $f_b$(RH=85%) versus $f$(RH=85%) at $\lambda$ =550 nm. Dots represent the mean value and error bars one standard deviation. The figure includes the results of a weighted bivariate fit according to York et al. (2004). The colors indicate the site type (dark green for mountain site, light green for rural, cyan for marine, dark blue for Arctic and black for urban sites), other studies are plot in gray. Those studies are: *SAG refers to Sagres (Portugal) with SAGP and SAGC refering to polluted and clean conditions, respectively (Carrico et al., 2000); CGO refers to Cape Grim (Australia) (Carrico et al., 1998); BND refers to Bondville (Illinois, USA) (Koloutsou-Vakakis et al., 2001); H1 and H2 are average values from two flights made off the west coast of the US (Hegg et al., 1996); CM, CP, CV and CD refer to marine, polluted, volcanic and dust dominated conditions during the ACE-Asia Experiment in the North Pacific Ocean (Carrico et al., 2003); and RA and HS refer to Regional S. Africa air and heavy smoke, respectively (Magi and Hobbs, 2003).

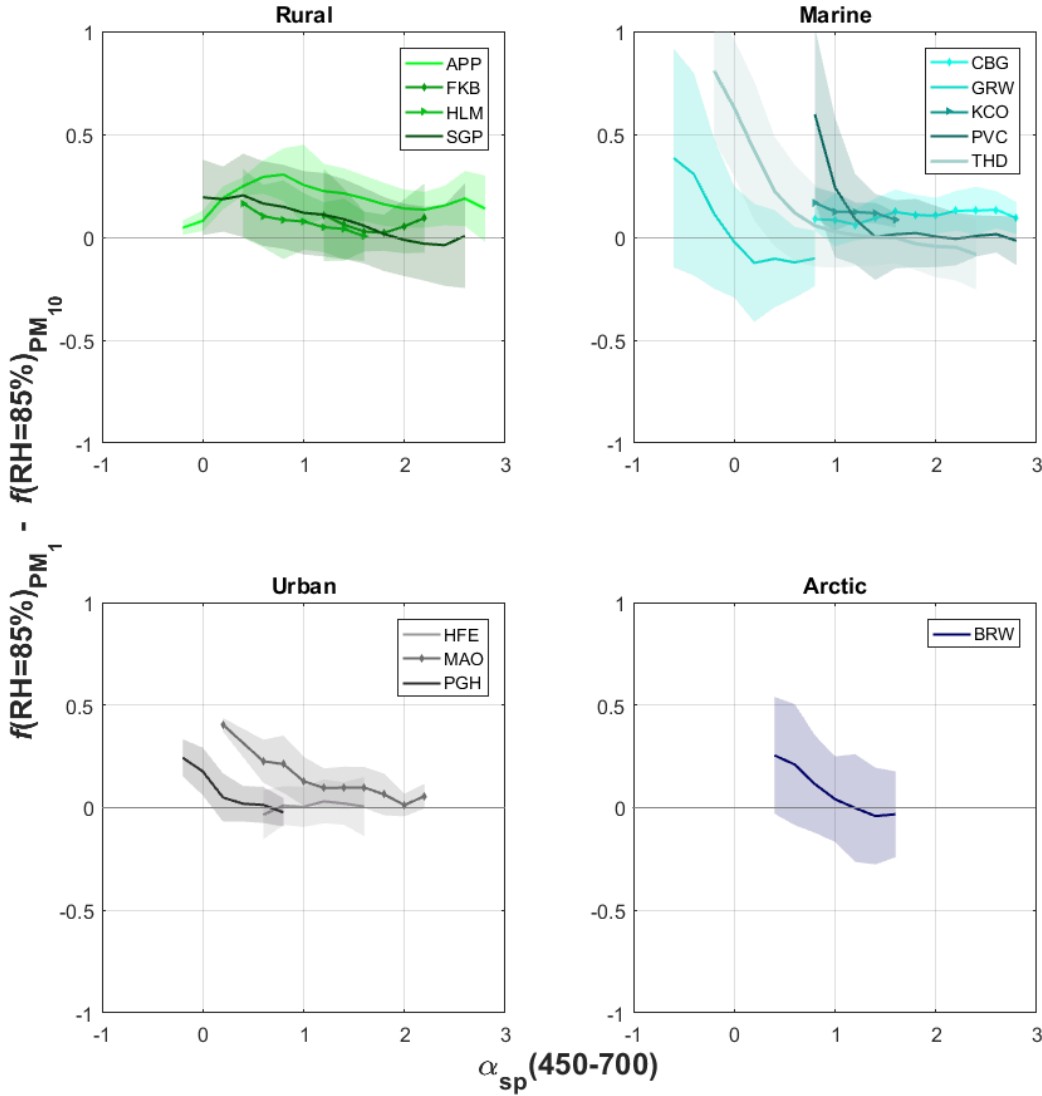

**Figure 3.** Difference in the $f(RH=85\%)$ in $PM_1$ and $PM_{10}$ size fractions at 550 nm as a function of the dry scattering Angström exponent, $\alpha_{sp}$, calculated between 700 and 450 nm. The solid line represents the mean value and the shaded region represents +/- 1 standard deviation. The points are binned in increments of 0.2 $\alpha_{sp}$. Only bins with standard error below 0.03 are considered.





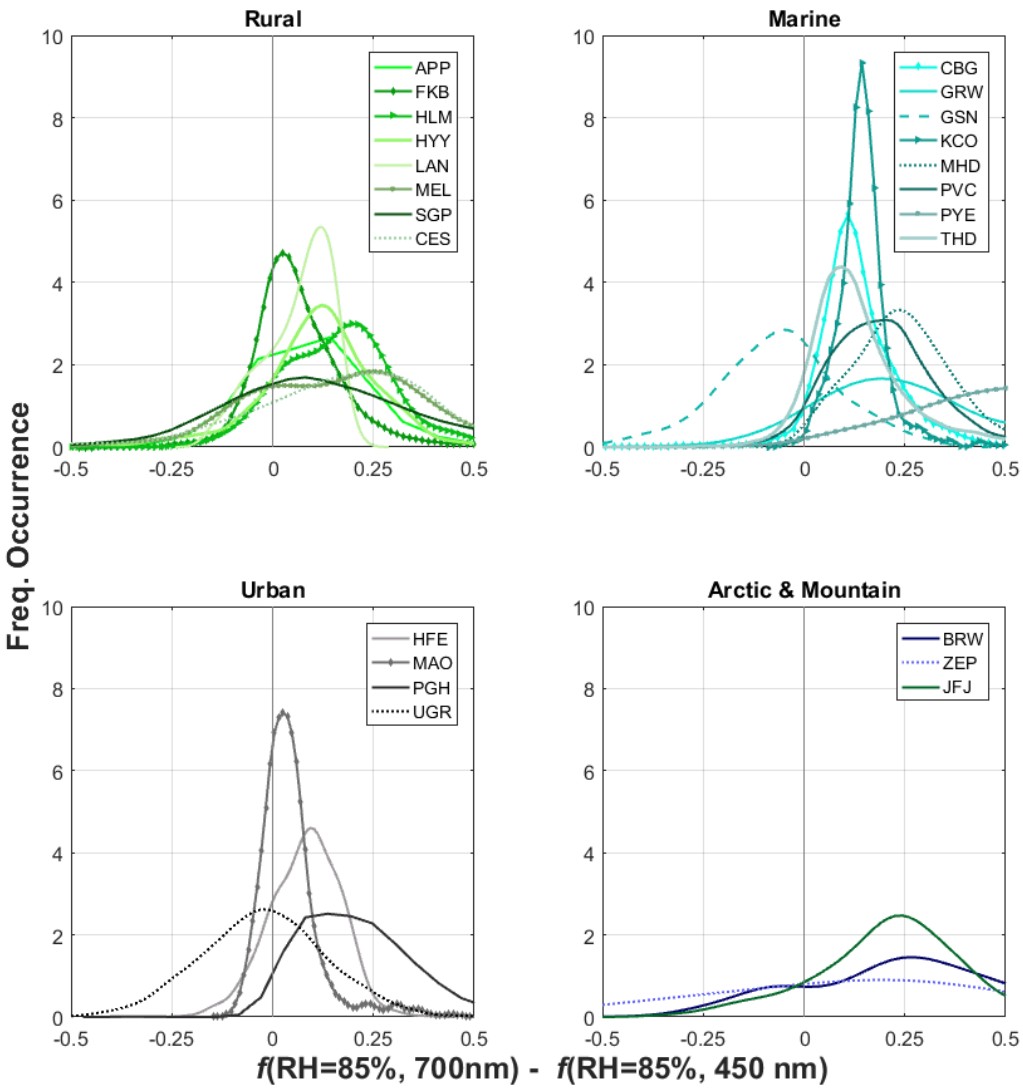

**Figure 4.** Frequency of occurrence of the difference in the $f$(RH=85%) at 700 nm and 450 nm.



**Figure 5.** Median values of single scattering albedo, backscatter fraction and ratio of the radiative forcing at certain RH to the radiative forcing at dry conditions (RH<40%). All variables refer to 550 nm wavelength





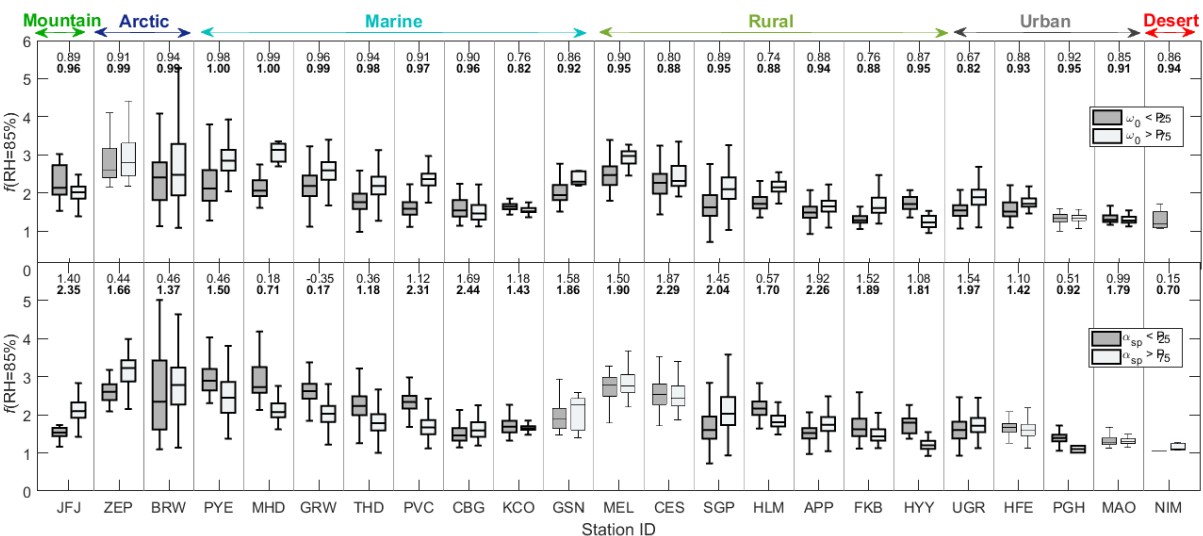

**Figure 6.** Boxplot of $f$(RH=85%) at 550 nm ($\lambda$ =525 nm at HYY) segregated by single scattering albedo, $\omega_0$, values (upper panel) and scattering Ångström exponent, $\alpha_{sp}$, values (lower panel). Sites are sorted by site type and decreasing $f$(RH=85%). For each box, the central mark is the median, the box extends vertically between the 25th and 75th percentiles, the whiskers extend to the most extreme data that are not considered outliers. The numbers on top indicate the 25th and 75th (in bold) percentiles used for segregating each dataset. Categories that are not significantly different are shown with thinner lines in the boxplot.





**Figure 7.** Scatter plot of mean $f$(RH=85%) versus single scattering albedo, $\omega_0$, at 550 nm ($\lambda$ =525 nm at HYY) color coded by the scattering Angström exponent, $\alpha_{sp}$ for the wavelength pair 450-700 nm at each site. The dots represent the mean value and the error bars are the standard deviation.