# Peer review of "A global study of hygroscopicity-driven light scattering enhancement in the context of other in-situ aerosol optical properties"

_Atmospheric Chemistry and Physics, 2020_

## Referee Comment (RC1) · Anonymous Referee #1 · 26 Apr 2021

Review of manuscript acp-2020-1250

GENERAL REMARKS

The article by Titos and co-workers presents a data rich and extensive analysis of aerosol light scattering properties at elevated relative humidity, based on observations from various sites on the globe. Covered aerosol types reach from polluted urban aerosol to aerosol at remote Arctic and mountain stations. Thus, the study presents an excellent overview of the influence of aerosol hygroscopic growth on the aerosol optical properties. One major target of the study was the attempt to deduce correlations between hygroscopic growth and resulting light scattering properties at ambient conditions and other aerosol properties which are measured more frequently than the more difficult-to-access hygroscopic growth behaviour. Although the study cannot recommend clear parameterisations of aerosol hygroscopic behaviour and their impact on the aerosol optical properties, it makes a substantial contribution to the highly relevant research question of the adequate representation of aerosol optical properties at ambient conditions in global models for the full variety of atmospheric aerosol types.

The paper is very well written and structured, and the number and quality of figures is adequate, with the exceptions of Figures 1 and 6 for which small changes are recommended. The study fits well into the scope of the journal and will be suitable for publication once few issues as discussed in the following have been considered.

SPECIFIC COMMENT

Some of the paragraphs in the Results and Discussions section are very descriptive. Adding a more in-depth physical interpretation of the observations would benefit the manuscript very much. Since these changes may require some work, the requested revisions are rated as Major.

1. Section 3.1 and description of Figure 1: The figure itself presents a huge data set but drawing conclusions from the present version is difficult. The authors state that the datasets from the different sites are not comparable since the sizes of the data sets and the covered seasons may differ significantly from site to site. Although it is almost impossible to present this large and diverse data set consistently, some improvements may be possible. One is to add the sizes of the datasets to Figure 1 (see also Minor Issue 6 below). In the description of the datasets the scatter of data is partially explained by the fact that, e.g., the marine sites may contain polluted as well as clean episodes, but how are "polluted" and clean" defined and why not separating the "polluted" from the "clean" air masses? By combining all datasets from one site to a single analysis, natural variability may cover correlations between air mass characteristics and aerosol hygroscopic growth. Furthermore, the explanation of Figure 1 is very descriptive without discussing physical properties (typical size distributions, chemical composition) which may explain the observed growth factors and may explain some of the observed agreements and disagreements between observations at different sites.

2. On line 325, it is mentioned that for rural stations there is no difference in hygroscopic growth observed between the size cuts of $PM_1$ and $PM_{10}$. However, looking at the bottom left panel of Figure 3, the dependence of $f(RH=85\%)_{PM1} - f(RH=85\%)_{PM10}$ shows an almost similar behaviour as for the other aerosol types, i.e., larger differences at smaller values of the scattering Ångström exponent which indicates enhanced presence of larger particles, and decreasing differences with increasing

scattering Ångström exponent values, which indicate the presence of a strong mode of small particles. It is recommended to revisit the discussion of Figure 3.

3. Likewise, the discussion of Figure 4 should be reworked. It is an interesting way of plotting the differences in humidity growth factors. As described in Section 2.1, f(RH, $\lambda$) values were determined from measurements of aerosol scattering coefficients with dry and humidified Integrating Nephelometers, using Equation (3). If this procedure is correctly understood, then the measurements include intrinsic size distribution information, since the particle size distribution has a direct impact on the aerosol scattering coefficient. It would be very helpful to see a discussion of physical properties in the description of Figure 4 which can explain the observations.

4. Section 3.3 on the Changes in RFE is very well written and explains very clearly the observed behaviour. It may serve as an example for the sections discussed above.

MINOR ISSUES

1. Please check the spelling of the term Ångström throughout the text to ensure consistent use. Sometimes it is written with a capital "A", sometimes with a capital "Å".

2. Equation (4) is misleading since the independent parameter of the Ångström exponents is a wavelength pair and not the difference between two wavelengths. It is recommended to write $\alpha_{sp}$ ($\lambda_1$, $\lambda_2$) instead of $\alpha_{sp}$ ($\lambda_1 - \lambda_2$).

3. Line 226: Rearrangement of the sentence is recommended for better understanding. Suggested change is "Figure 1 shows the dry total scattering (top plot) and f(RH=85%) statistics (bottom plot) for individual sites …".

4. Line 232 ff: The authors mention their results suggest "that aerosol loading does not control the magnitude of f(RH=85%)". But how should that be the case since hygroscopic growth depends on the composition of the particles and not on their total load? I suggest adding a short explanation here why this dependence is not observed or remove the statement.

5. Line 436: Suggested rephrasing: the authors may write "This is consistent with the hypothesis that …" instead of "This is consistent with the idea that … ". There are some other sentences in the manuscript, where a more precise language is recommended.

6. Figure 1: it is mentioned in the text that the sizes of the data sets may differ largely between the different sites. Adding the size of the datasets to the individual box and whiskers per site would certainly help interpreting the figure adequately.

7. Figure 6: The numbers of single-scattering albedo (top panel) and scattering Ångström exponent are difficult to read. The same holds for the numbers on the figure legend. Enhancement of numbers is suggested.

---

## Referee Comment (RC2) · Anonymous Referee #2 · 29 Apr 2021

The submission by Tito et al provides an extensive compilation of measurements of the aerosol humidification factor "f(RH)" for total scattering and hemispheric backscattering for 23 sites representing a broad range of aerosol types. Most of the scattering measurements were obtained with TSI integrating nephelometers at three nominal red, green, and blue wavelengths. In addition, more than half of the sites included alternating size-cut operation permitting Tito et al to probe the humidification effects as a function of fine or coarse aerosol contributions. And while the humidification of absorption was not explicitly addressed (and is assumed to be negligible), concurrent dry aerosol absorption measurements were frequently available which permitted

Tito et al to explore correlations between between (dry) SSA and f(RH) as well as relationships between f(RH) and wavelength dependence, backscatter fraction, and fine/coarse contributions for potential use in estimating f(RH) when humidified measurements are unavailable. Ultimately, while completely general relationships remain elusive, the authors demonstrate 1) agreement with previous reported published results and 2) relationships with limited utility when applied to different "types" of locations, for example marine, rural, urban, arctic, etc.

I recommend accepting this paper for publication after addressing several minor concerns. Although there are quite a few changes I've suggested, in my opinion they are not very arduous.

I've attached commented version of the submission as "acp-2020-1250-manuscript-version2.RC2.pdf" .

1. Line 54, suggest "smaller" rather than "lower". (Just personal preference, of course.)

2. line 58-60, suggest slight reordering of text in this paragraph.

3. line 90, poor hypenation

4. line 103, awkward phrasing. Simplify?

5. line 109-110: A double parenthetical? Revise or strike.

6. line 125, hyphenate "scattering-related" perhaps?

7. line 144, more detail on selection criteria please

8. line 164, Important to identify the specific ARM datastreams as there are multiple and they are not interchangeable.

9. Eq 4, recommend alternate more intuitive form.

10. line 191, substitude "vary" for "range" to avoid unwanted connotation of "wavelength range"

11. line 235, Can't generalize from a sample size of 2.

12. line 244, Well said.

13. line 286-7, rephrase for clarity

14. line 288, Unneeded comma.

15. line 290, Cleaner?

16. lines 312-14, Suggest deeper explanation.

17. lines 374-381, figure 5. Focus this figure and discussion on SSA.

18. lines 452-467, figure 7. Suggest revising fig 7 to let symbol size represent angstrom exponent and then let the symbol color indicate the predominate aerosol type (rural,urban, marine, etc.) This may also open the door to more a more illuminating discussion than provided by the current figure.

Please also note the supplement to this comment:
https://acp.copernicus.org/preprints/acp-2020-1250/acp-2020-1250-RC2-supplement.pdf

**Supplement:**

[revised manuscript text omitted]

---

## Author Comment (AC1) · 24 Jun 2021

Dear editor and reviewers,

We thank both reviewers for their review and valuable comments, which have helped to improve our manuscript. A detailed point-by-point response to the issues raised by both referees is provided in the attached file.

Please let us know if you have any further questions.

Please also note the supplement to this comment:
https://acp.copernicus.org/preprints/acp-2020-1250/acp-2020-1250-AC1-supplement.pdf

---

## Author Response (AR1)

**Manuscript by Titos et al. "A global study of hygroscopicity-driven light scattering enhancement in the context of other in-situ aerosol optical properties" - Reply to reviewers**

We thank the reviewers for their valuable and positive comments that have helped to improve the quality of our manuscript. The reviewers' comments are in *italicized black font*. Our replies are given below in blue, and when we refer to text that has been changed in the manuscript (main text and supplement materials) we show it in this reply letter in red (these correspond to the changes in the manuscript and supplement which are also shown in red), while original text is shown in black.

**1 Reviewer 1**

*GENERAL REMARKS*

*The article by Titos and co-workers presents a data rich and extensive analysis of aerosol light scattering properties at elevated relative humidity, based on observations from various sites on the globe. Covered aerosol types reach from polluted urban aerosol to aerosol at remote Arctic and mountain stations. Thus, the study presents an excellent overview of the influence of aerosol hygroscopic growth on the aerosol optical properties. One major target of the study was the attempt to deduce correlations between hygroscopic growth and resulting light scattering properties at ambient conditions and other aerosol properties which are measured more frequently than the more difficult-to-access hygroscopic growth behaviour. Although the study cannot recommend clear parameterisations of aerosol hygroscopic behaviour and their impact on the aerosol optical properties, it makes a substantial contribution to the highly relevant research question of the adequate representation of aerosol optical properties at ambient conditions in global models for the full variety of atmospheric aerosol types. The paper is very well written and structured, and the number and quality of figures is adequate, with the exceptions of Figures 1 and 6 for which small changes are recommended. The study fits well into the scope of the journal and will be suitable for publication once few issues as discussed in the following have been considered.*

*SPECIFIC COMMENTS*

*Some of the paragraphs in the Results and Discussions section are very descriptive. Adding a more in-depth physical interpretation of the observations would benefit the manuscript very much. Since these changes may require some work, the requested revisions are rated as Major.*

*1. Section 3.1 and description of Figure 1: The figure itself presents a huge data set but drawing conclusions from the present version is difficult. The authors state that the datasets from the different sites are not comparable since the sizes of the data sets and the covered seasons may differ significantly from site to site. Although it is almost impossible to present this large and*

*diverse data set consistently, some improvements may be possible. One is to add the sizes of the datasets to Figure 1 (see also Minor Issue 6 below). In the description of the datasets the scatter of data is partially explained by the fact that, e.g., the marine sites may contain polluted as well as clean episodes, but how are "polluted" and "clean" defined and why not separating the "polluted" from the "clean" air masses? By combining all datasets from one site to a single analysis, natural variability may cover correlations between air mass characteristics and aerosol hygroscopic growth. Furthermore,the explanation of Figure 1 is very descriptive without discussing physical properties (typical size distributions, chemical composition) which may explain the observed growth factors and may explain some of the observed agreements and disagreements between observations at different sites.*

We agree that it would be enlightening to have further information about characteristics of the aerosol (e.g., size distribution/chemistry) or atmospheric conditions (e.g., polluted vs clean). These types of information would definitely contribute to better understanding the $f$(RH) variability among sites. However, there are several reasons why we could not include such details in this paper: (1) the desired aerosol information is not readily available for many of the sites. For example, many of the DOE sites did not make aerosol size distribution or aerosol chemistry measurements. (2) When the size distribution data are available they may need further review before being used. As a recent example of such effort in this direction, Rose et al. (2021) presents a comprehensive analysis of particle size distribution for one year (2016 or 2017) at 39 stations. Only 4 of the stations included in Rose et al. (2021) have $f$(RH) measurements but the measurement period differs, making the results less comparable. (3) A key physical characteristic is the size distribution of the coarse aerosol, but measuring that is difficult (Pfeifer et al., 2016) and most sites do not have that available (especially for longer periods). (4) Some sites make aerosol chemistry measurements but the types of measurements vary widely (daily or weekly filters analyzed for common ions to more detailed mass spectrometer measurements). Further, many of the chemical measurements only cover the sub-$\mu$m size - again information on the coarse mode is lacking. (5) Finally most sites do not have clean and polluted sectors defined or readily accessible flags identifying those sectors. Wind direction is not a perfect metric for identifying clean vs polluted except perhaps for very local sources. We (Burgos et al., 2019) spent several years getting the $f$(RH) data into an useable and consistent form. Because the $f$(RH) dataset included dry scattering coefficients data with spectral resolution we could calculate a rough metric for size (i.e., $\alpha_{sp}$) from data we had already thoroughly assessed. The composition proxy (i.e., $\omega_0$) was derived from quality controlled aerosol absorption coefficient. In particular, absorption coefficients were obtained from EBAS or DOE/ARM databases. Those datasets have been previously used in global phenomenologies of aerosol optical properties (i.e., Collaud Coen et al. (2020) and Zanatta et al. (2016)). The parameters $\alpha_{sp}$ and $\omega_0$ provide some indication of aerosol predominant size distribution and composition (e.g., Figure 6 in the originally submitted paper), but going beyond that is best left to station mentors with their additional data and an understanding of their particular site characteristics. Comparison with the existing literature is also difficult, since for many sites there are no publications on aerosol size distribution and/or chemistry, and when it exists the time periods may differ and the way the information is presented may not be ideal for comparison with our study. Anyway, where possible we have included citations about aerosol physical/chemical characteristics and their possible interpretation of the $f$(RH) measurements presented in this manuscript. In addition, the following statements has been added:

60    Lines 273-276: Detailed information about the aerosol size distribution and chemical composition would be needed to better understand the observed differences amongst sites. Section 3.4 further explores this variability using $\alpha_{sp}$ and $\omega_0$ as qualitative indicators of predominant aerosol size and composition due to the lack of concurrent size distribution and chemical composition measurements at most sites

Lines 164-166: The data availability (measurement period and number of valid datapoints) for each site can be found in Burgos
65    et al. (2019), their Figure 1 and Tables 5 and 6. Table S1 of the supplementary material includes the number of available $f$(RH=85%) and $f_b$(RH=85%) values.

Lines 284-285: The number of available $f$(RH=85%) and $f_b$(RH=85%) values is included in Table S1 and in Tables 5 and 6 of Burgos et al. (2019).

*2. On line 325, it is mentioned that for rural stations there is no difference in hygroscopic growth observed between the size cuts*
70    *of PM1 and PM10. However, looking at the bottom left panel of Figure 3, the dependence of f(RH=85%)PM1–f(RH=85%)PM10 shows an almost similar behaviour as for the other aerosol types, i.e., larger differences at smaller values of the scattering Ångström exponent which indicates enhanced presence of larger particles, and decreasing differences with increasing scattering Ångström exponent values, which indicate the presence of a strong mode of small particles. It is recommended to revisit the discussion of Figure 3.*

75    For rural stations, there is almost no change in the $f$(RH) PM1-PM10 difference with $\alpha_{sp}$. We believe that the reviewer meant "urban sites" instead of "rural sites" since this is what corresponds with bottom left panel of Figure 3. For urban sites, we agree with the reviewer: MAO and PGH show a decrease in the $f$(RH) PM1-PM10 with $\alpha_{sp}$, while no dependence is observed for HFE. This observed dependence for urban sites is lower than for the marine sites. While the three sites are characterized as "urban", the three sites are located in different environments (China, India and Brazil) with very different emission sources
80    and atmospheric processes. The anthropogenic urban emissions are likely to be different across these three sites, specially concerning the types of fuel burned and burning conditions. At PGH, Dumka et al. (2017) observed higher $f$(RH) values for lower $\alpha_{sp}$ and this was associated with biomass burning events. According to Dumka et al. (2017), coarse carbonaceous aerosols predominated during winter leading to lower $f$(RH). It is important to note that in PGH, $\alpha_{sp}$ values were always <1 indicating predominance of coarse particles, so the distinction between coarse or fine particles predominance cannot be well
85    addressed at this site with the $\alpha_{sp}$ parameter. At MAO, Almeida et al. (2019) reported that when fine particles predominate the aerosol tends to be less soluble which could led to lower $f$(RH). The study of Almeida et al. (2019) is not directly comparable with ours, since these authors focused on the fine mode fraction. Therefore, more study is needed with additional information from the individual sites about composition, size and sources to explain the observed behaviour at urban sites. As stated in our previous comment, this detail of analysis is out of the scope of this paper. The following information have been included in the
90    revised version of this manuscript:

Lines 345-355: A similar, but less marked, dependence of $f$(RH=85%) $PM_1$-$PM_{10}$ separation with $\alpha_{sp}$ is observed at two of the three urban sites investigated here. The anthropogenic urban emissions are likely to be different across these three sites, specially concerning the types of fuel burned and burning conditions. At PGH, Dumka et al. (2017) observed higher $f$(RH) values for lower $\alpha_{sp}$ and this was associated with biomass burning events. According to Dumka et al. (2017), coarse

95 carbonaceous aerosols predominated during winter leading to lower $f$(RH). It is important to note that in PGH $\alpha_{\mathrm{sp}}$ values were always <1 (see Fig. 3) indicating predominance of coarse particles, so the distinction between coarse or fine particles predominance cannot be well addressed at this site with the $\alpha_{\mathrm{sp}}$ parameter. At MAO, Almeida et al. (2019) reported that when fine particles predominate the aerosol tends to be less soluble which could lead to lower $f$(RH). The study of Almeida et al. (2019) is not directly comparable with ours, since these authors focused on the fine mode fraction. Therefore, more study

100 is needed with additional information from the individual sites about composition, size and sources to explain the observed behaviour at urban sites.

*3. Likewise, the discussion of Figure 4 should be reworked. It is an interesting way of plotting the differences in humidity growth factors. As described in Section 2.1, f(RH,lamda) values were determined from measurements of aerosol scattering coefficients with dry and humidified Integrating Nephelometers, using Equation (3). If this procedure is correctly understood,*

105 *then the measurements include intrinsic size distribution information, since the particle size distribution has a direct impact on the aerosol scattering coefficient. It would be very helpful to see a discussion of physical properties in the description of Figure 4 which can explain the observations.*

We agree with the reviewer that a discussion of concurrent measurements of aerosol physical properties (i.e. size distribution) would definitely help to better understand the observed differences in the $f$(RH) spectral dependency. However, as noted in our

110 response to Comment 1, that detail of information is not available for most of the sites analyzed in this study. Where possible, we have tried to link Figure 4 and additional aerosol information (size or chemistry) from the literature.

*4. Section 3.3 on the Changes in RFE is very well written and explains very clearly the observed behaviour. It may serve as an example for the sections discussed above*

We thank the reviewer for his/her positive comments.

115 *MINOR ISSUES*

*1. Please check the spelling of the term Ångström throughout the text to ensure consistent use. Sometimes it is written with a capital "A", sometimes with a capital "Å".*

We have checked the spelling and now the term Ångström is written consistently throughout the manuscript.

*2. Equation (4) is misleading since the independent parameter of the Ångström exponents is a wavelength pair and not the*

120 *difference between two wavelengths. It is recommended to write αsp(lamda1,lamda2) instead of αsp(lamda1-lamda2)*

We agree with the reviewer, this change has been implemented in the new version of the manuscript.

*3. Line 226: Rearrangement of the sentence is recommended for better understanding.Suggested change is "Figure 1 shows the dry total scattering (top plot) and f(RH=85%) statistics(bottom plot)for individual sites ...".*

Done.

125 *4. Line 232 ff: The authors mention their results suggest "that aerosol loading does not control the magnitude of f(RH=85%)". But how should that be the case since hygroscopic growth depends on the composition of the particles and not on their total load? I suggest adding a short explanation here why this dependence is not observed or remove the statement.*

We agree with the reviewer, RH scattering enhancement depends on particle size and composition and not on aerosol concentration. However, there are some situations where this trend may exist. There are definitely places where changes in loading

130 correspond to changes in other aerosol properties (e.g., $\omega_0$ or $\alpha_{\mathrm{sp}}$) suggesting that loading can be related to aerosol composition. For example, Delene and Ogren (2002) and Andrews et al. (2011) show that systematic variability can be observed between loading and $\omega_0$ or loading and $\alpha_{\mathrm{sp}}$. In particular, at urban sites higher scattering coefficients may be indicative of higher aerosol load associated with pollution events, or at marine sites a higher aerosol load may be due to higher influence of anthropogenic pollution at the site. At some mountain sites higher loading may correspond to an incursion of dust aerosol.

135 However, as shown in Fig. 1 of the manuscript there are no trends with aerosol load and $f(\mathrm{RH})$. We have added the following sentences to clarify why we expected a possible loading effect:

Lines 239-243: The $f(\mathrm{RH}=85\%)$ values shown are for measurements made for total or $\mathrm{PM}_{10}$ aerosol. The sites in Fig. 1 are grouped by their assumed dominant aerosol type (e.g., marine, rural, urban, etc.). Since previous research has shown systematic variability between loading and proxies for aerosol size and composition at individual sites (e.g., Delene and Ogren (2002),

140 their Figures 8 and 9), within the groupings, the $f(\mathrm{RH}=85\%)$ values are ordered by the aerosol loading (using the dry aerosol scattering coefficient as a proxy for aerosol amount).

Lines 247-250: The lack of dependence of $f(\mathrm{RH})$ on aerosol loading amongst site types is likely due to sites experiencing different aerosol sources/types throughout the measurement period which can not be disentangled when looking at the overall statistics of $f(\mathrm{RH})$ and loading. This is looked at in more detail in Section 3.4.

145 *5. Line 436: Suggested rephrasing: the authors may write "This is consistent with the hypothesis that ..." instead of "This is consistent with the idea that... ".There are some other sentences in the manuscript, where a more precise language is recommended.*

We have changed this sentence and improved the language throughout the manuscript

*6. Figure1: it is mentioned in the text that the sizes of the data sets may differ largely between the different sites. Adding the*
150 *size of the datasets to the individual box and whiskers per site would certainly help interpreting the figure adequately.*

Since the size of the data sets for $f(\mathrm{RH}=85\%)$ and $f_b(\mathrm{RH}=85\%)$ are different, adding this information in the graph may be misleading. Therefore, we have included this information in a Table in the revised Supplementary Material (Table S1). Note that this information was already available in Burgos et al. (2019), their Tables 5 and 6 and Figure 1.

*7. Figure 6: The numbers of single-scattering albedo (top panel) and scattering Ångström exponent are difficult to read. The*
155 *same holds for the numbers on the figure legend. Enhancement of numbers is suggested.*

Done, we have enhanced the font size.

**2 Reviewer 2**

*The submission by Titos et al provides an extensive compilation of measurements of the aerosol humidification factor "f(RH)" for total scattering and hemispheric backscattering for 23 sites representing a broad range of aerosol types. Most of the scattering measurements were obtained with TSI integrating nephelometers at three nominal red, green, and blue wavelengths. In addition, more than half of the sites included alternating size-cut operation permitting Titos et al to probe the humidification effects as a function of fine or coarse aerosol contributions. And while the humidification of absorption was not explicitly addressed (and is assumed to be negligible), concurrent dry aerosol absorption measurements were frequently available which permitted Titos et al to explore correlations between (dry) SSA and f(RH) as well as relationships between f(RH) and wavelength dependence, backscatter fraction, and fine/coarse contributions for potential use in estimating f(RH) when humidified measurements are unavailable. Ultimately, while completely general relationships remain elusive, the authors demonstrate 1) agreement with previous reported published results and 2) relationships with limited utility when applied to different "types" of locations, for example marine, rural, urban, arctic, etc.I recommend accepting this paper for publication after addressing several minor concerns. Although there are quite a few changes I've suggested, in my opinion they are not very arduous.*

*1. Line 54, suggest "smaller" rather than "lower". (Just personal preference, of course.)*

We agree, done.

*2. line 58-60, suggest slight reordering of text in this paragraph.*

We have reordered this paragraph according to the reviewer's suggestions.

*3. line 90, poor hypenation*

We have avoided automatic hyphenation throughout the manuscript

*4. line 103, awkward phrasing. Simplify?*

In order to simplify this sentence, we have split it in two and modified them accordingly.

*5. line 109-110: A double parenthetical? Revise or strike.*

We have revised this sentence avoiding double parenthesis

*6. line 125, hyphenate "scattering-related" perhaps?*

Done

*7. line 144, more detail on selection criteria please*

The criteria applied in Burgos et al. (2019) to generate the $f$(RH=85%) data used in this study consists in: first, only those humidograms spanning a RH range larger than $30\%$ in the humidified nephelometer are considered; second, goodness-of-fit criterion is applied such that humidogram fits with a R-squared value less than 0.5 are also flagged as invalid. A stricter goodness of fit requirement is used for Hyytiälä and Jungfraujoch (R-squared value threshold was set to 0.7 and 0.8, respectively) where higher variability is observed in the RH scans. This information has been included in the new version of the manuscript.

*8. line 164, Important to identify the specific ARM datastreams as there are multiple and they are not interchangeable.*

We thank the reviewer for pointing this out. For the DOE/AMF deployments the datastreams used here for absorption coefficient and calculated $\omega_0$ are the AIPAVG1OGREN.c1. This information has been included in Section 2.2: Dry aerosol optical property dataset and in Code and Data availability section.

*9. Eq 4, recommend alternate more intuitive form.*

We have modified it as follows:

$$\alpha_{\mathrm{sp}}(\lambda_1, \lambda_2) = -\frac{\log(\sigma_{\mathrm{sp}}(\lambda_1)/\sigma_{\mathrm{sp}}(\lambda_2))}{\log(\lambda_1/\lambda_2)} \tag{1}$$

*10. line 191, substitude "vary" for "range" to avoid unwanted connotation of "wavelength range"*

Done

*11. line 235, Can't generalize from a sample size of 2.*

We agree with the reviewer, we have modified this paragraph as follows:

Lines 250-251: Urban and a dust-dominated site tend to show the lowest hygroscopicity based on $f$(RH=85%) values, while the two Arctic sites (ZEP and BRW) show high $f$(RH=85%) values

*12. line 244, Well said.*

Thanks.

*13. line 286-7, rephrase for clarity*

Done

*14. line 288, Unneeded comma.*

Agree, it has been removed

*15. line 290, Cleaner?*

Yes, we have removed "value of".

*16. lines 312-14, Suggest deeper explanation. Perhaps a more satisfying/penetrating explanation would reference the relative change in the cross-sectional area rather than particle diameter since to first order scattering and water uptake will both be proportional to area. Suppose for example you have one large droplet and a collection of identically-sized smaller droplets of the same composition where the cross-sectional area of the large droplet is equal to the total combined cross-sectional area of the smaller droplets. In the approximation of geometric optics the single larger droplet will have approximately the same scattering as the collection of smaller droplets. Meanwhile under the same RH they will also have the same total surface areas and therefore approximately the same water uptake in terms of volume or mass. The uptake of the same volume of water by the large droplet will results in a smaller change in cross-sectional area for the larger droplet than for the collection of smaller droplets. Therefore, the collection of smaller droplets will exhibit a larger increase in scattering. Note that this argument doesn't need to invoke different inherent hygroscopicity, which as you correctly note also plays a role to the extent that dust (as a dominant component of coarse aerosols in non-marine locations) is only weakly hygroscopic. In addition, the dilution effect on hydrophilic nitrates and sulfates will also favor stronger hygroscopicity for the smaller droplets than for the larger (presumably more aqueous) droplets.*

We acknowledge the reviewer's suggestion. We focused the discussion in terms of particle diameter because typically growth factors are estimated as a function of particle diameter and the scattering efficiency is referred to the size parameter (relationship between the wavelength of incident light and particle size). However, we agree that the original statement was hard to read. Therefore, we have replaced the original sentence "This results in smaller particles with smaller diameter change due to water uptake exhibiting higher scattering enhancement due to a larger increase in scattering efficiency than bigger particles with more diameter growth but with almost no change in the scattering efficiency."

with the following:

Lines 331-334: This may result in smaller particles having less diameter change due to water uptake (lower hygrosocopicity) but more scattering enhancement due to a larger relative increase in scattering efficiency; this is in contrast to larger particles which may have more diameter growth (higher hygrosocopicity) but exhibit only a little relative change in scattering efficiency.

*17. lines 374-381, figure 5. Focus this figure and discussion on SSA. To be honest, I think this figure should be simplified by eliminating both the SSA and b response and focusing only on the change in forcing. Since absorption is assumed constant, any change in SSA must be due only to the hygroscopic growth of the droplets, and we already know the particles GROW in response to increasing RH so certainly the scattering and thus the SSA must also increase. Likewise, it is clear as the particle size increases the forward scattered lobe increases faster than the backscattering so certainly the backscatter fraction must decrease with increasing RH. And we even know that the forcing will increase with RH, so the only remaining element of interest is how the forcing changes for the different regimes (rural, urban, marine, arctic/mountain==cold?) compared to one another. So you can simplify the figure to focus on those comparisons and contrast.*

We thank the reviewer for his/her suggestion, however, we have decided to keep the discussion on $\omega_0$, $b$ and RFE. Although we have neglected the RH absorption enhancement in this calculation, this figure highlights the ample range of variations observed in the $\omega_0$ enhancement for the different site types. It shows that not all the sites exhibit the same $\omega_0$-$RH$ behaviour, resulting in varying RFE enhancement at elevated RH. For example, CES and APP exhibit very similar trends of $b$ with RH, but CES exhibits a stronger increase of $\omega_0$ with RH than is observed at APP. This difference results in a larger increase in the forcing efficiency due to RH at CES, the site which shows the largest enhancement in the forcing efficiency at high RH (4-fold) relative to dry conditions.

*18. lines 452-467, figure 7. Suggest revising fig 7 to let symbol size represent angstrom exponent and then let the symbol colour indicate the predominate aerosol type (rural,urban, marine, etc.) This may also open the door to more a more illuminating discussion than provided by the current figure.*

Thank you for this suggestion, we have modified Figure 7 of the manuscript in order to include information on the site type as recommended by the reviewer. However, we believe that $\alpha_{sp}$ in the colour scale is easier to interpret than in the dots size scale, and allows comparison of this figure with Fig. S4 that shows the relationship between $f$(RH), $\omega_0$ and $\alpha_{sp}$ at the individual sites. Consequently, in the revised version of the manuscript we have replaced Figure 7 by Figure R1 that includes the site type information as a symbol (see legend). Despite this additional information it is still difficult to infer a pattern as a function of site type or $\alpha_{sp}$. This lack of pattern with respect to site type has been noted in the revised version of the manuscript. In addition, the data show in this new Figure differs slightly from what it was shown in the original submission. In the current Figure, only

concurrent measurements of $f(\text{RH})$, $\omega_0$ and $\alpha_{\text{sp}}$ have been used while in the previous version all the available data for each variable was included.

[Figure]

**Figure R1.** Scatter plot of mean $f(\text{RH}=85\%)$ versus single scattering albedo, $\omega_0$, at 550 nm ($\lambda =$525 nm at HYY) color coded by Ångström exponent, $\alpha_{\text{sp}}$ for the wavelength pair 450-700 nm at each site. The error bars are the standard deviation.

**References**

Almeida, G. P., Bittencourt, A. T., Evangelista, M. S., Vieira-Filho, M. S., and Fornaro, A.: Characterization of aerosol chemical composition from urban pollution in Brazil and its possible impacts on the aerosol hygroscopicity and size distribution, Atmospheric Environment, 202, 149 – 159, https://doi.org/https://doi.org/10.1016/j.atmosenv.2019.01.024, http://www.sciencedirect.com/science/article/pii/S1352231019300469, 2019.

Andrews, E., Ogren, J., Bonasoni, P., Marinoni, A., Cuevas, E., Rodrigues, S., Sun, J., Jaffe, D., Fischer, E., Baltensperger, U., Weingartner, E., Collaud Coen, M., Sharma, S., Macdonald, A., Leaitch, W., Lin, N.-H., Laj, P., Arsov, T., Kalapov, I., Jefferson, A., and Sheridan, P.: Climatology of aerosol radiative properties in the free troposphere, Atmos. Res., 102, 365–393, 2011.

Burgos, M., Andrews, E., Titos, G., Alados-Arboledas, L., Baltensperger, U., Day, D., Jefferson, A., Kalivitis, N., Mihalopoulos, N., Sherman, J., Sun, J., Weingartner, E., and Zieger, P.: A global view on the effect of water uptake on aerosol particle light scattering, Scientific Data, 6, https://doi.org/10.1038/s41597-019-0158-7, 2019.

Collaud Coen, M., Andrews, E., Alastuey, A., Arsov, T. P., Backman, J., Brem, B. T., Bukowiecki, N., Couret, C., Eleftheriadis, K., Flentje, H., Fiebig, M., Gysel-Beer, M., Hand, J. L., Hoffer, A., Hooda, R., Hueglin, C., Joubert, W., Keywood, M., Kim, J. E., Kim, S.-W., Labuschagne, C., Lin, N.-H., Lin, Y., Lund Myhre, C., Luoma, K., Lyamani, H., Marinoni, A., Mayol-Bracero, O. L., Mihalopoulos, N., Pandolfi, M., Prats, N., Prenni, A. J., Putaud, J.-P., Ries, L., Reisen, F., Sellegri, K., Sharma, S., Sheridan, P., Sherman, J. P., Sun, J., Titos, G., Torres, E., Tuch, T., Weller, R., Wiedensohler, A., Zieger, P., and Laj, P.: Multidecadal trend analysis of in situ aerosol radiative properties around the world, Atmospheric Chemistry and Physics, 20, 8867–8908, https://doi.org/10.5194/acp-20-8867-2020, 2020.

Delene, D. J. and Ogren, J. A.: Variability of Aerosol Optical Properties at Four North American Surface Monitoring Sites, Journal of the Atmospheric Sciences, 59, 1135–1150, https://doi.org/10.1175/1520-0469(2002)059<1135:VOAOPA>2.0.CO;2, 2002.

Dumka, U., Kaskaoutis, D., Sagar, R., Chen, J., Singh, N., and Tiwari, S.: First results from light scattering enhancement factor over central Indian Himalayas during GVAX campaign, Science of the Total Environment, 605, 124–138, 2017.

Pfeifer, S., Müller, T., Weinhold, K., Zikova, N., Dos Santos, S., Marinoni, A., Bischof, O., Kykal, C., Ries, L., Meinhardt, F., Aalto, P., Mihalopoulos, N., and Wiedensohler, A.: Intercomparison of 15 aerodynamic particle size spectrometers (APS 3321): Uncertainties in particle sizing and number size distribution, Atmospheric Measurement Techniques, 9, 1545–1551, https://doi.org/10.5194/amt-9-1545-2016, 2016.

Rose, C., Collaud Coen, M., Andrews, E., Lin, Y., Bossert, I., Lund Myhre, C., Tuch, T., Wiedensohler, A., Fiebig, M., Aalto, P., Alastuey, A., Alonso-Blanco, E., Andrade, M., Artíñano, B., Arsov, T., Baltensperger, U., Bastian, S., Bath, O., Beukes, J. P., Brem, B. T., Bukowiecki, N., Casquero-Vera, J. A., Conil, S., Eleftheriadis, K., Favez, O., Flentje, H., Gini, M. I., Gómez-Moreno, F. J., Gysel-Beer, M., Hallar, A. G., Kalapov, I., Kalivitis, N., Kasper-Giebl, A., Keywood, M., Kim, J. E., Kim, S.-W., Kristensson, A., Kulmala, M., Lihavainen, H., Lin, N.-H., Lyamani, H., Marinoni, A., Martins Dos Santos, S., Mayol-Bracero, O. L., Meinhardt, F., Merkel, M., Metzger, J.-M., Mihalopoulos, N., Ondracek, J., Pandolfi, M., Pérez, N., Petäjä, T., Petit, J.-E., Picard, D., Pichon, J.-M., Pont, V., Putaud, J.-P., Reisen, F., Sellegri, K., Sharma, S., Schauer, G., Sheridan, P., Sherman, J. P., Schwerin, A., Sohmer, R., Sorribas, M., Sun, J., Tulet, P., Vakkari, V., van Zyl, P. G., Velarde, F., Villani, P., Vratolis, S., Wagner, Z., Wang, S.-H., Weinhold, K., Weller, R., Yela, M., Zdimal, V., and Laj, P.: Seasonality of the particle number concentration and size distribution: a global analysis retrieved from the network of Global Atmosphere Watch (GAW) near-surface observatories, Atmospheric Chemistry and Physics Discussions, 2021, 1–69, https://doi.org/10.5194/acp-2020-1311, https://acp.copernicus.org/preprints/acp-2020-1311/, 2021.

295   Zanatta, M., Gysel, M., Bukowiecki, N., Müller, T., Weingartner, E., Areskoug, H., Fiebig, M., Yttri, K., Mihalopoulos, N., Kouvarakis, G., Beddows, D., Harrison, R., Cavalli, F., Putaud, J., Spindler, G., Wiedensohler, A., Alastuey, A., Pandolfi, M., Sellegri, K., Swietlicki, E., Jaffrezo, J., Baltensperger, U., and Laj, P.: A European aerosol phenomenology-5: Climatology of black carbon optical properties at 9 regional background sites across Europe, Atmospheric Environment, 145, 346 – 364, https://doi.org/https://doi.org/10.1016/j.atmosenv.2016.09.035, http://www.sciencedirect.com/science/article/pii/S135223101630735X, 2016.